# SAMA: Towards Multi-Turn Referential Grounded Video Chat with Large Language Models

Ye Sun[1], Hao Zhang[2], Henghui Ding[1][†], Tiehua Zhang[3], Xingjun Ma[1][†], Yu-Gang Jiang[1]

[1]Fudan University, [2]HKUST, [3]Tongji University
https://github.com/sunye23/SAMA

## Abstract

Achieving fine-grained spatio-temporal understanding in videos remains a major challenge for current Video Large Multimodal Models (Video LMMs). Addressing this challenge requires mastering two core capabilities: video referring understanding, which captures the semantics of video regions, and video grounding, which segments object regions based on natural language descriptions. However, most existing approaches tackle these tasks in isolation, limiting progress toward unified, referentially grounded video interaction. We identify a key bottleneck in the lack of high-quality, unified video instruction data and a comprehensive benchmark for evaluating referentially grounded video chat. To address these challenges, we contribute in three core aspects: dataset, model, and benchmark. First, we introduce SAMA-239K, a large-scale dataset comprising 15K videos specifically curated to enable joint learning of video referring understanding, grounding, and multi-turn video chat. Second, we propose the SAMA model, which incorporates a versatile spatio-temporal context aggregator and a Segment Anything Model to jointly enhance fine-grained video comprehension and precise grounding capabilities. Finally, we establish SAMA-Bench, a meticulously designed benchmark consisting of 5,067 questions from 522 videos, to comprehensively evaluate the integrated capabilities of Video LMMs in multi-turn, spatio-temporal referring understanding and grounded dialogue. Extensive experiments and benchmarking results show that SAMA not only achieves strong performance on SAMA-Bench but also sets a new state-of-the-art on general grounding benchmarks, while maintaining highly competitive performance on standard visual understanding benchmarks.

## 1 Introduction

Recent years have witnessed remarkable progress in Large Multimodal Models (LMMs) [46, 57], driving significant advances in general vision-language understanding [37, 75, 31, 13]. However, extending these capabilities effectively to the video domain remains a critical and open research challenge [35, 69, 42, 32]. Unlike static images, videos introduce additional complexity through temporal dynamics, requiring models not only to perform holistic scene comprehension but also to achieve fine-grained alignment between language and temporally evolving objects and actions within a continuous spatio-temporal context [2]. While current Video LMMs [35, 69, 42, 32, 30, 33] demonstrate strong performance in global scene understanding, they often fall short in fine-grained spatio-temporal reasoning. This limitation is particularly evident when models are required to interpret references to specific entities and deliver precisely grounded information amid complex and dynamic activities—capabilities that are essential for applications such as detailed event forensics, interactive robotic assistance, and nuanced content analysis.

---

[†]Corresponding authors: henghui.ding@gmail.com, xingjunma@fudan.edu.cn

39th Conference on Neural Information Processing Systems (NeurIPS 2025).

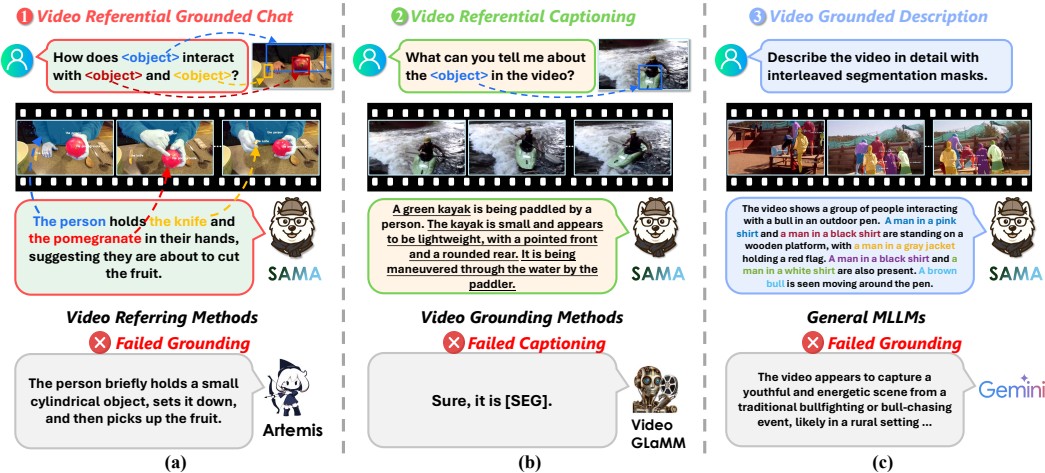

Figure 1: Comparisons with prior specialized and general MLLMs. Our SAMA model excels in multiple fine-grained video understanding tasks that require both referring and grounding capabilities: complex video referential grounded chat, video referential captioning, and video grounded description.

Enabling fine-grained video understanding hinges on mastering two fundamental and deeply interconnected capabilities: **video referring understanding**—the ability to comprehend semantics associated with designated video regions [48, 68, 20]—and **video grounding**—the capacity to accurately segment specific video regions based on natural language descriptions [12, 9, 67, 45, 1, 64]. In the image domain, substantial progress has been made with multimodal large language models (MLLMs) such as Ferret [65], Shikra [3], GPT4RoI [72], LLaVA-Grounding [70], and CogVLM [59], which integrate object-level representations into MLLMs to enable coordinate output or segmentation masks for fine-grained image comprehension. However, analogous advances in the video domain remain limited and face unique challenges. For video referring understanding, recent works like Artemis [48], VideoRefer [68], and OmniRGPT [20] demonstrate reasonable regional understanding by leveraging object prompts. Meanwhile, in video grounding, specialized LLM-based models such as VISA [61], VideoLISA [1], and VideoGLaMM [45] have been developed to segment target objects based on textual descriptions. Yet, a critical limitation persists: most of these efforts treat referring understanding and grounding as distinct tasks, neglecting their intrinsic interdependence—namely, that both fundamentally require precise spatio-temporal alignment with semantic cues. This separation prevents current Video LMMs from achieving a unified, human-like capability to both comprehend nuanced object details and fluidly interpret dynamic scenes, as shown in Figure 1.

We identify a key bottleneck hindering further progress: the lack of high-quality, unified video instruction data specifically tailored to enable the joint learning of referring understanding and grounding, along with the absence of a comprehensive benchmark for evaluating integrated performance [68]. While existing video understanding datasets typically focus on isolated tasks such as action recognition or captioning [22, 27], and foundational grounding datasets offer valuable segmentation or localization annotations for visual entities [9, 11, 53, 44, 74, 10, 14], a critical gap remains: the lack of resources that combine these components to support advanced referential grounded video dialogue. Specifically, current datasets fall short in providing: **1)** rich, multi-turn referring-based video dialogues; **2)** precise spatio-temporal grounding of objects and actions mentioned in those dialogues; and **3)** explicitly structured joint supervision to facilitate the simultaneous learning of referring understanding and grounding within an interactive framework.

To address these challenges, we contribute in 3 key aspects: *dataset, model, and benchmark.*

● *Dataset.* We present **SAMA-239K**, a large-scale video instruction-tuning dataset specifically curated to facilitate the joint learning of video referring understanding and grounding. To construct this dataset, we developed a LLM-based automated annotation pipeline that first enriches video data by generating accurate segmentation masks using HQ-SAM [24]. Building upon these grounding annotations and leveraging advanced prompt engineering with Gemini-1.5 Pro [55], we generate 67K detailed object-level descriptions and 172K multi-turn conversational QA pairs to support training. Unlike previous single-turn video QA datasets, SAMA-239K is designed to emulate natural human

interaction, evolving from basic observations to complex reasoning and contextual inference, while explicitly linking conversational entities to their corresponding spatio-temporal masks.

• *Model.* We propose **SAMA**, a novel Video LMM architecture that seamlessly integrates fine-grained spatial and temporal understanding. SAMA introduces a versatile spatio-temporal aggregator to capture video-level temporal dynamics, alongside a powerful SAM2 [50] for extracting frame-level pixel-level semantics. This unified design enables the model to fully exploit the rich supervision in SAMA-239K, supporting end-to-end learning across video referring understanding, video grounding, and multi-turn referential video dialogue within a single cohesive framework.

• *Benchmark.* We introduce SAMA-Bench, a comprehensive benchmark comprising 522 diverse videos curated from four established datasets, totaling 5,067 challenging questions. It consists of two main components: SAMA-Bench$^G$, which evaluates referential grounded dialogue through a broad range of question types involving object behaviors, interactions, attributes, and spatial relationships; and SAMA-Bench$^C$, which assesses video referring region captioning by requiring models to generate detailed captions for specified object categories, including persons, animals, tools, and vehicles.

• *Evaluation.* Extensive benchmarking and experiments show that our SAMA model outperforms existing grounding LMMs on SAMA-Bench and also achieves strong results on established grounding benchmarks. To the best of our knowledge, SAMA is the first Video LMM to successfully unify fine-grained referential understanding and grounded dialogue, achieving state-of-the-art performance on both image and video referring segmentation tasks.

## 2 Related Work

**Multimodal Large Language Models.** The remarkable capabilities of LLMs [46, 57, 41] across a wide range of language-centric tasks have reshaped the landscape of artificial intelligence (AI), establishing a powerful foundation for extending reasoning abilities to visual understanding. In response, research has rapidly converged on the development of image-based MLLMs. Pioneering works such as LLaVA [37], MiniGPT-4 [75], and InstructBLIP [8] have demonstrated impressive visual understanding by effectively integrating visual inputs with LLMs. Building upon these successes in processing static images, attention has increasingly shifted toward Video LMMs [33, 35, 69, 42, 30]. To capture the rich spatio-temporal information in videos, Video LMMs typically extract sequential visual features from frames using pre-trained vision backbones. These features are often processed via direct token concatenation [5, 35], and then interleaved with textual embeddings as input to the LLM to generate responses. Despite promising results on tasks requiring holistic understanding—such as general video question answering and captioning—existing Video LMMs still face notable challenges in fine-grained, spatio-temporal object-level grounding and understanding [65, 3, 59, 70, 15].

**MLLMs for Referring and Grounding.** Enabling fine-grained interaction in multimodal systems requires MLLMs to accurately refer to and ground specific entities within visual inputs. In the image domain, notable progress has been made. One major direction, represented by models such as Kosmos-2 [47], Shikra [3], Ferret [65], GPT4RoI [72], and CogVLM [59], equips MLLMs with the ability to process and generate spatial location information, typically in the form of bounding boxes or point coordinates expressed numerically or as discrete tokens. Another active direction integrates segmentation capabilities, enabling models to produce pixel-level masks aligned with textual references. This is often achieved by connecting LLMs with segmentation models like SAM [26] through specialized integration mechanisms [70, 28, 49, 56].

Extending referring and grounding capabilities to the video domain introduces new challenges, particularly due to temporal complexity and the need for consistent object tracking. Recent works like Artemis [48], VideoRefer [68], Omni-RGPT [20], and DAM [34] have advanced fine-grained video understanding through region-level descriptions conditioned on prompts. However, they do not support dynamic segmentation mask generation within conversational outputs. In contrast, VideoGLaMM [45] achieves strong referring segmentation in videos but focuses on grounded descriptions, lacking support for interactive, fine-grained video dialogue. This highlights a key gap: the lack of models that jointly support precise object-level grounding and nuanced referring in interactive video conversations. We bridge this gap through improved datasets, model designs, and benchmarks, achieving rich, grounded, and interactive video dialogue.

# 3 SAMA

## 3.1 SAMA-239K Data Creation

As illustrated in Figure 2, our data construction leverages a sophisticated LLM-based automated annotation pipeline, centered around Gemini-1.5 Pro [55], which systematically transforms existing video referring segmentation or grounding datasets into rich, grounded conversational data.

**Object-level Description Generation.** Our initial aim is to obtain precise, object-centric textual descriptions. For grounding datasets that only offer

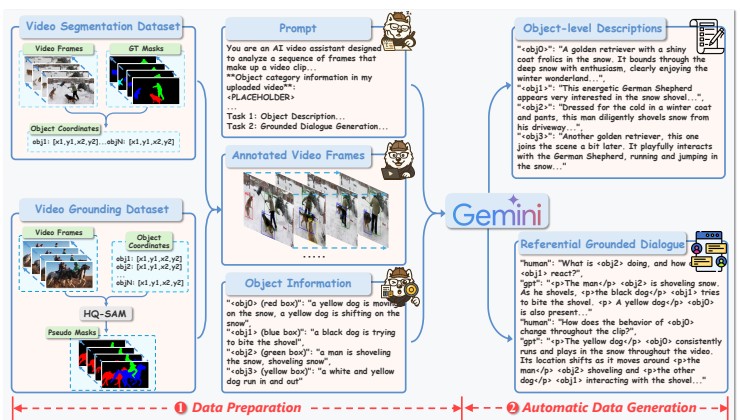

Figure 2: Data creation pipeline of SAMA-239K. Best viewed in zoom.

bounding boxes but lack pixel-level masks, we first employ HQ-SAM [24] to generate high-quality pseudo masks from the existing box information. Inspired by SoM [62], we then leverage segmentation mask information to highlight objects of interest within sampled video frames using distinctly colored bounding boxes. These visually cued frames, along with relevant categorical or action information for the highlighted objects, are provided to Gemini to generate fine-grained, descriptive captions for each identified object instance.

**Referential Grounded Dialogue Generation.** Building upon the rich, localized descriptions, we further task Gemini with creating multi-turn question-answer pairs. These dialogues are designed to emulate natural human conversational patterns, initiating with basic observational questions about one or more objects and progressively advancing to more complex reasoning that might involve inter-object relationships or inferential steps. A critical instruction is that the generated answers must explicitly incorporate grounding information, linking back to the previously established spatio-temporal masks of the referred entities.

Finally, through an automated filtering process to ensure the correctness of the generated data format, our proposed SAMA-239K has 172,296 referential grounded video-QA triplets along with 67,005 object-level descriptions in total.

## 3.2 SAMA-239K Data Characteristics

**Spatio-Temporal Dynamics.** For understanding complex object motion and achieving precise, temporally-consistent grounding, we select training data from referring video object segmentation datasets, primarily MeViS [9] and Ref-YouTube-VOS [53], which provide video sequences with dense annotations of objects undergoing significant motion changes.

**Large Vocabulary Knowledge.** To ensure SAMA comprehends a broad range of objects and their diverse textual descriptions, we incorporate LV-VIS [58], a video instance segmentation dataset that contributes a wide vocabulary of object categories, vital for open-world generalization.

**Complex Scene Robustness.** Real-world video understanding demands robustness against complex environmental conditions. To cultivate this, we incorporate data from SAV [50], which contributes challenging scenarios featuring occlusions and motion blur. Furthermore, we utilize video grounding dataset VidSTG [74] to ensure the model maintain consistent tracking and contextual comprehension over long temporal horizons.

## 3.3 SAMA Model

Here, we present the architecture of SAMA, which builds upon Sa2VA [67] and introduces novel contributions to synergize fine-grained frame-level information with long-range temporal context.

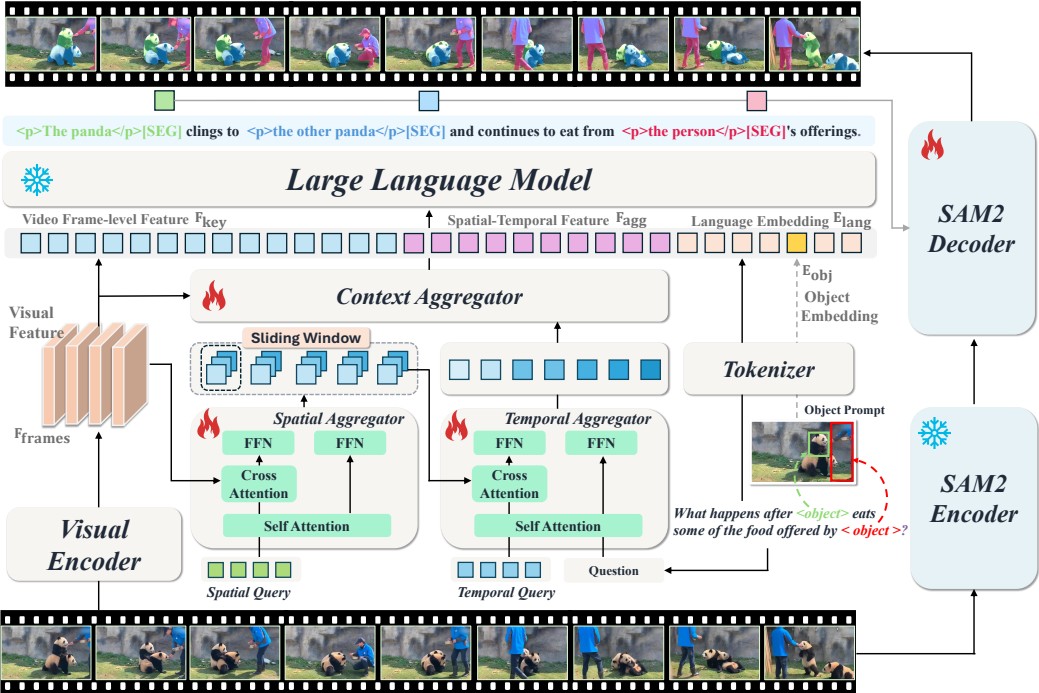

Figure 3: Model architecture of our SAMA for referential grounded video understanding.

### 3.3.1 Overall Architecture

The overall architecture of the SAMA model is illustrated in Figure 3. Conventional Video LMMs typically process video input via two main strategies: (1) concatenating tokenized features from sparsely sampled keyframes [5, 67, 35], or (2) aggressively pooling or merging features from individual frames into a very small number of tokens to accommodate longer sequences [33, 19, 51, 71]. We argue that both approaches have inherent limitations: the former often overlooks critical long-range temporal dynamics, while the latter tends to lose the fine-grained visual detail necessary for accurate referring segmentation and detailed object understanding.

SAMA is designed to overcome these limitations by effectively integrating both rich, frame-level visual information and robust, long-range temporal context. As depicted, a video input first passes through a Visual Encoder (e.g., a pre-trained ViT [5]) to extract per-frame visual features $\mathbf{F}_{\text{frames}} \in \mathbb{R}^{N \times P \times D_v}$, where $N$ is the number of sampled frames, $P$ is the number of patches per frame, and $D_v$ is the feature dimension. A subset of these frame features, typically from keyframes, $\mathbf{F}_{\text{key}} \subset \mathbf{F}_{\text{frames}}$, are directly flattened and projected to serve as fine-grained visual tokens for the LLM. Concurrently, features from a more extensive set of frames $\mathbf{F}_{\text{long}} \subseteq \mathbf{F}_{\text{frames}}$ (potentially all $N$ frames) are processed by our proposed Versatile Spatial-Temporal-Context Aggregator (detailed in Sec. 3.3.2) to derive a compact yet informative representation of long-range temporal dynamics, denoted as $\mathbf{F}_{\text{agg}}$.

The language input is processed by a tokenizer to produce language embeddings $\mathbf{E}_{\text{lang}}$. For referring understanding tasks involving specific objects, we incorporate object embeddings $\mathbf{E}_{\text{obj}}$ into $\mathbf{E}_{\text{lang}}$. Specifically, $\mathbf{E}_{\text{obj}}$ is obtained by performing mask pooling on the visual features $\mathbf{F}_{\text{key}}$ of the relevant frame using a bounding box prompt for the target object. $\mathbf{E}_{\text{obj}}$ is then inserted into the language embeddings $\mathbf{E}_{\text{lang}}$ at the position corresponding to the object reference.

Finally, $\mathbf{F}_{\text{key}}$, $\mathbf{F}_{\text{agg}}$, and $[\mathbf{E}_{\text{lang}}, \mathbf{E}_{\text{obj}}]$ are concatenated and fed into a large language model $\mathbf{\Phi}_{\text{LLM}}$ to generate the textual response. For grounding tasks, if the LLM outputs a special [SEG] token, the hidden state corresponding to this token $\mathbf{H}_{\text{SEG}}$ is passed to the decoder of a pre-trained SAM2 model [50] to produce the final segmentation mask $\mathbf{M}_{\text{obj}}$ for the referred object.

### 3.3.2 Versatile Spatial-Temporal-Context Aggregator

To efficiently capture both spatial details and temporal evolution from a potentially large number of video frames $\mathbf{F}_{\text{long}}$, we introduce a three-stage aggregator module.

**Spatial Aggregator.** Given $N_L$ frames selected for long-range temporal modeling, where each frame $i$ has visual features $\mathbf{F}_{\text{long}}^{(i)} \in \mathbb{R}^{P \times D_v}$, the initial goal of the spatial aggregator is to condense the spatial information from each frame while retaining essential semantics, thereby reducing computational load for subsequent temporal modeling. Inspired by Q-Former [31], for each frame $i$, we employ a set of $K_S$ learnable spatial queries $\mathbf{Q}_S \in \mathbb{R}^{K_S \times D_v}$. These queries interact with the frame's patch features $\mathbf{F}_{\text{long}}^{(i)}$ through cross-attention mechanisms, followed by feed-forward networks (FFN), to produce a compact representation $\mathbf{Z}_S^{(i)} \in \mathbb{R}^{K_S \times D_v}$:

$$\mathbf{Z}_S^{(i)} = \text{SpatialAgg}(\mathbf{Q}_S, \mathbf{F}_{\text{long}}^{(i)}). \tag{1}$$

In our implementation, we use $K_S = 32$ tokens per frame. This results in a sequence of spatially aggregated tokens $\mathbf{Z}_S = [\mathbf{Z}_S^{(1)}, \mathbf{Z}_S^{(2)}, ..., \mathbf{Z}_S^{(N_L)}] \in \mathbb{R}^{(N_L \cdot K_S) \times D_v}$.

**Temporal Aggregator.** To capture comprehensive spatio-temporal information relevant to the input query, the sequence of spatially aggregated tokens $\mathbf{Z}_S$ is processed by the temporal aggregator. We employ a sliding window approach: tokens from $\mathbf{Z}_S$ corresponding to a window of $W_T$ frames are iteratively fed into a temporal Q-Former structure. The temporal aggregator takes $K_T$ learnable temporal queries $\mathbf{Q}_T \in \mathbb{R}^{K_T \times D_v}$ and interacts with the windowed spatial tokens $\mathbf{Z}_{S,\text{window}}$. Crucially, the textual question embedding $\mathbf{E}_{\text{question}}$ (and any associated object embeddings $\mathbf{E}_{\text{obj}}$ if present within the question context) is also injected into this temporal Q-Former by concatenation with the temporal queries, which guides the aggregation process to focus on temporal dynamics relevant to the query:

$$\mathbf{Z}_T^{\text{window}} = \text{TemporalAgg}(\mathbf{Q}_T, \mathbf{Z}_{S,\text{window}}, \mathbf{E}_{\text{question}}, [\mathbf{E}_{\text{obj}}]). \tag{2}$$

The outputs from each sliding window are then concatenated to yield a query-aware spatio-temporal representation $\mathbf{Z}_T \in \mathbb{R}^{K_{final} \times D_v}$, where $K_{final}$ is the resulting number of temporal tokens.

**Context Aggregator.** Finally, to ensure that the spatio-temporal information $\mathbf{Z}_T$ is effectively aligned and integrated with the original fine-grained visual features $\mathbf{F}_{\text{long}}$, we employ a context aggregator. This module uses the frame features $\mathbf{F}_{\text{long}}$ as queries, and the learned query-aware temporal features $\mathbf{Z}_T$ as keys and values in an attention mechanism, allowing the model to selectively enhance the frame representations with relevant long-range temporal context. Subsequently, we apply adaptive average pooling to the spatio-temporally enhanced frame features, compressing each frame into a single token. These tokens are then projected through a linear layer to align with the language embedding space of the LLM. The final aggregated context feature $\mathbf{F}_{\text{agg}}$ is obtained through the following process:

$$\mathbf{F}_{\text{agg}} = \text{Mean}(\text{Softmax}((\mathbf{F}_{\text{long}}\mathbf{W}^Q) \times (\mathbf{Z}_T\mathbf{W}^K)^T / \sqrt{C}) \times \mathbf{Z}_T\mathbf{W}^V)\mathbf{W}^P, \mathbf{F}_{\text{agg}} \in \mathbb{R}^{N \times D_v}, \tag{3}$$

where $\mathbf{W}^Q, \mathbf{W}^K, \mathbf{W}^V, \mathbf{W}^P$ denote the linear projection layer. $\mathbf{F}_{\text{agg}}$, combined with the fine-grained keyframe features and language features are fed into the LLM to generate the target response.

### 3.4 SAMA-Bench

To evaluate the integrated referential understanding and grounded dialogue capabilities of Video LMMs, we propose SAMA-Bench, constructed using the same annotation pipeline as SAMA-239K. SAMA-Bench comprises 5,067 questions synthesized from 522 videos across four public validation datasets: MeViS [9], Ref-YouTube-VOS [53], LVVIS [58], and VidSTG [74].

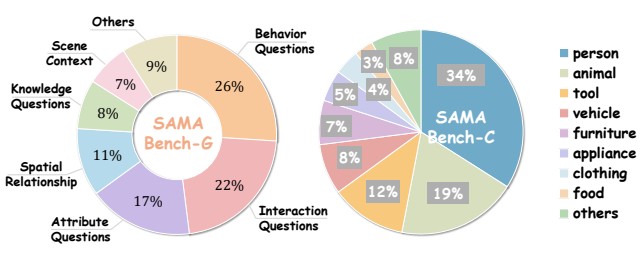

(a) Question types in **Bench-G**  (b) Category list in **Bench-C**

Figure 4: Data characteristics of SAMA-Bench.

It includes two sub-benchmarks: SAMA-Bench$^{\text{G}}$ (3,036 questions) and SAMA-Bench$^{\text{C}}$ (2,031 questions). The distributions of question types and object categories are shown in Figure 4.

Table 1: Performance comparisons on SAMA-Bench. The best results are **boldfaced**, and second-best results are underlined. "–" denotes that the model does not support the specified output format. Entries in gray represent that the original model is incapable of performing the task. Values in red show SAMA's performance change relative to corresponding Sa2VA variants.

| Method | SAMA-Bench$^G$ | | | | | SAMA-Bench$^C$ | | |
|---|---|---|---|---|---|---|---|---|
| | mIoU | Recall | METEOR | CIDEr | CLAIR | METEOR | CIDEr | CLAIR |
| *Generalist Models* | | | | | | | | |
| InternVL2.5-26B [5] | – | – | 0.14 | 0.33 | 0.47 | 0.09 | 0.07 | 0.31 |
| Gemini-2.0 Flash [54] | – | – | 0.11 | 0.24 | 0.53 | 0.11 | 0.16 | 0.52 |
| Gemini-1.5 Pro [54] | – | – | 0.15 | 0.48 | **0.62** | 0.13 | 0.27 | **0.56** |
| *Specialist Models* | | | | | | | | |
| *Image-level models* | | | | | | | | |
| GLaMM [49] + SAM2 [50] | 0.28 | 0.04 | 0.04 | 0.03 | 0.16 | 0.04 | 0.02 | 0.33 |
| Shikra [3] + SAM2 [50] | 0.27 | 0.26 | 0.08 | 0.15 | 0.32 | 0.04 | 0.01 | 0.29 |
| Ferret-7B [65] + SAM2 [50] | 0.64 | 0.44 | 0.14 | 0.21 | 0.37 | 0.10 | 0.12 | 0.31 |
| Ferret-13B [65] + SAM2 [50] | 0.64 | 0.43 | 0.14 | 0.20 | 0.39 | 0.11 | 0.10 | 0.31 |
| *Video-level models* | | | | | | | | |
| Sa2VA-1B [67] | 0.09 | 0.07 | 0.10 | 0.16 | 0.31 | 0.06 | 0.03 | 0.26 |
| Sa2VA-4B [67] | 0.55 | 0.25 | 0.05 | 0.02 | 0.19 | 0.00 | 0.00 | 0.07 |
| Sa2VA-8B [67] | 0.64 | 0.17 | 0.02 | 0.02 | 0.20 | 0.00 | 0.00 | 0.13 |
| **SAMA-1B** | 0.67 (0.58↑) | 0.53 (0.46↑) | 0.16 (0.06↑) | 0.56 (0.40↑) | 0.53 (0.22↑) | **0.14** (0.08↑) | 0.31 (0.28↑) | 0.45 (0.19↑) |
| **SAMA-4B** | 0.69 (0.14↑) | **0.55** (0.30↑) | 0.17 (0.12↑) | 0.65 (0.63↑) | 0.57 (0.38↑) | 0.13 (0.13↑) | 0.30 (0.30↑) | 0.48 (0.41↑) |
| **SAMA-8B** | **0.70** (0.06↑) | **0.55** (0.38↑) | **0.17** (0.15↑) | **0.69** (0.67↑) | 0.58 (0.38↑) | 0.13 (0.13↑) | **0.32** (0.32↑) | 0.50 (0.37↑) |

**SAMA-Bench$^G$** This sub-benchmark evaluates a model's ability to conduct fine-grained, referential, and grounded visual dialogue. It focuses on region-level understanding and responses that require precise spatio-temporal grounding. As illustrated in Figure 4 (a), question types include **object behaviors**, **interactions**, **attributes**, **spatial relationships**, and more, requiring identification and reasoning over grounded entities. Following GLaMM [49, 45], we assess grounding performance using mIoU and Recall, and dialogue quality using METEOR, CIDEr, and CLAIR.

**SAMA-Bench$^C$** This sub-benchmark focuses on generating accurate descriptions for specified spatio-temporal video regions. As shown in Figure 4 (b), the benchmark spans diverse object categories, including **person**, **animal**, **tool**, **vehicle**, **furniture**, **appliance**, **clothing**, and **food**. Given a first-frame prompt, models are required to generate detailed captions for a target region. Description quality is evaluated using METEOR, CIDEr, and CLAIR.

## 4 Experiments

### 4.1 Experimental Setup

**Training Datasets.** SAMA is trained on a diverse set of image/video QA and referring segmentation/grounding datasets, including LLaVA-1.5 665K [38], the ChatUniVi [21] dataset, image-based referring/grounding data (refCOCO/+/g [23, 43], GRand-F [49]), and video referring segmentation datasets (Ref-YouTube-VOS [53], MeViS [9], and ReVOS [61]). Critically, training is enhanced with our proposed SAMA-239K dataset, comprising 239K instances of referential grounded dialogue and object-level descriptions.

**Implementation Details.** We implement SAMA leveraging the XTuner [7] codebase. Model training and inference is conducted on 8 NVIDIA A100 GPUs (80GB). During the instruction tuning phase, we make the parameters of our spatial-temporal-context aggregator and the decoder of the SAM2 model [50] trainable to learn spatial-temporal information and inject referential grounded video chat capability into the base model. The model weights are initialized using the pre-trained Sa2VA [67] to benefit from its existing grounding capabilities. The initial learning rate is set to 4e-5. The maximum sequence length for the LLM is configured to 8,192.

**Evaluation Metrics.** For SAMA-Bench, grounding performance on SAMA-Bench$^G$ is measured using mIoU and Recall. Text generation quality on both SAMA-Bench$^G$ and SAMA-Bench$^C$ is evaluated using METEOR, CIDEr, and CLAIR. For standard image/video referring segmentation tasks, we adopt widely used metrics such as cIoU and J&F. Chat performance is assessed with VLMEvalKit [16].

### 4.2 Main Results

**Performance Comparisons on SAMA-Bench.** Table 1 presents a comprehensive performance comparison on SAMA-Bench, encompassing both the SAMA-Bench$^G$ and SAMA-Bench$^C$ sub-benchmarks. To ensure fair evaluation across diverse model types, tailored protocols were applied. For image-level specialist models (e.g., GLaMM [49], Shikra [3], Ferret [65]) that accept coordinate

Table 2: Performance comparisons on referring segmentation in images and videos. **Bold** and underlined values indicate the best and second-best results, respectively. Red highlights SAMA's performance difference from corresponding Sa2VA variants.

| Method | Image Segmentation | | | | Video Segmentation | | | |
|---|---|---|---|---|---|---|---|---|
| | RefCOCO [23] | RefCOCO+ [23] | RefCOCOg [66] | GCG [49] | MeViS [9] | Ref-DAVIS17 [25] | Ref-YTVOS [53] | ReVOS [61] |
| *Image-level models* | | | | | | | | |
| LISA-7B [28] | 74.1 | 62.4 | 66.4 | – | – | – | – | – |
| PixelLM-7B [52] | 73.0 | 66.3 | 69.3 | – | – | – | – | – |
| GLaMM-7B [49] | 79.5 | 72.6 | 74.2 | 28.9 | – | – | – | – |
| LLaVA-G-7B [70] | 77.1 | 68.8 | 71.5 | – | – | – | – | – |
| GSVA-13B [60] | 79.2 | 70.3 | 75.7 | – | – | – | – | – |
| OMG-LLaVA-7B [73] | 78.0 | 69.1 | 72.9 | 29.9 | – | – | – | – |
| *Video-level models* | | | | | | | | |
| VideoGLaMM [45] | – | – | – | – | 45.15 | 69.5 | – | – |
| VISA-13B [61] | 72.4 | 59.8 | 65.5 | – | 44.5 | 70.4 | 63.0 | 50.9 |
| VideoLISA-3.8B [1] | 73.8 | 63.4 | 68.3 | – | 44.4 | 68.8 | 63.7 | 47.5 |
| Sa2VA-4B [67] | 82.4 | 77.6 | 79.7 | 31.0 | 46.4 | 73.7 | 71.3 | 54.1 |
| Sa2VA-8B [67] | 82.6 | 78.0 | 80.3 | 32.2 | 51.5 | 75.9 | 72.3 | 57.6 |
| **SAMA-4B** | 82.5 (0.1↑) | 77.9 (0.3↑) | 80.3 (0.6↑) | 32.6 (1.6↑) | 48.3 (1.9↑) | 74.1 (0.4↑) | 71.5 (0.2↑) | 58.8 (4.7↑) |
| **SAMA-8B** | 82.7 (0.1↑) | 78.1 (0.1↑) | 80.6 (0.3↑) | 31.7 (0.5↓) | 53.7 (2.2↑) | 77.3 (1.4↑) | 72.8 (0.5↑) | 59.1 (1.5↑) |

Table 3: Performance comparisons on chat benchmarks in images and videos. **Bold** and underlined values indicate the best and second-best results, respectively.

| Method | Image Chat | | | | | | Video Chat |
|---|---|---|---|---|---|---|---|
| | MME [17] | MMBench [39] | SEED-Bench [29] | AI2D [29] | MMStar [4] | SQA$^{test}$ [40] | Video-MME [18] |
| *Generalist MLLMs* | | | | | | | |
| LLAVA-1.5-13B [36] | 1531 | 68.8 | 70.1 | - | - | - | - |
| Video-LLaVA-7B [35] | - | 60.9 | - | - | - | - | 39.9 |
| LLaMA-VID-7B [33] | 1521 | 65.1 | 59.9 | - | - | - | - |
| mPLUG-Owl3-8B [63] | - | 77.6 | - | - | - | - | 53.5 |
| InternVL2-8B [6] | - | 81.7 | 76.2 | - | - | - | 54.0 |
| *MLLMs with segmentation capability* | | | | | | | |
| *Image-level models* | | | | | | | |
| PixelLM-7B [52] | 309/135 | 17.4 | - | - | - | - | - |
| GLaMM-7B [49] | 14/9 | 36.8 | - | - | - | - | - |
| OMG-LLaVA-7B [73] | 1177/235 | 47.9 | 56.5 | - | - | - | - |
| *Video-level models* | | | | | | | |
| Sa2VA-8B [67] | **1690**/610 | **84.4** | **76.5** | **82.7** | **62.4** | **97.4** | **54.3** |
| **SAMA-8B** | 1639/**621** | 80.8 | 76.2 | 79.4 | 60.1 | 95.3 | 51.8 |

inputs but not direct video, we adopted a multi-step evaluation. First, these models were queried with questions containing target object coordinates from the initial video frame. Their coordinate-inclusive textual responses were then processed by Gemini to validate object correspondence. Finally, these validated coordinates prompted SAM2 [50] to generate segmentation masks for the video. For generalist and video models lacking explicit coordinate input (e.g., Gemini, InternVL, Sa2VA), visual cues like colored bounding boxes were overlaid on frames for querying their referential capabilities. The results in Table 1 highlight two key findings: **(1)** SAMA consistently outperforms all baselines on both sub-benchmarks. Notably, SAMA-8B achieves an mIoU of 0.70 and a Recall of 0.55, significantly surpassing Sa2VA, Shikra, and GLaMM, and exceeding the strong Ferret-13B + SAM2 baseline by 6% mIoU and 12% Recall. **(2)** SAMA also demonstrates strong text generation capabilities. It achieves the highest METEOR and CIDEr scores among all models, even outperforming Gemini-1.5 Pro, and significantly surpassing Ferret. These results underscore SAMA's superior spatio-temporal reasoning and fine-grained grounding abilities.

**Performance Comparisons on Referring Segmentation in Images and Videos.** The results in Table 2 demonstrate SAMA's robust and consistent performance across both image and video referring segmentation tasks. **(1)** SAMA achieves superior image segmentation performance. Specifically, SAMA-8B obtains top scores on RefCOCO (82.7), RefCOCO+ (78.1), and RefCOCOg (80.6), outperforming all previous models, including GLaMM-7B and VISA-13B. Furthermore, on the GCG benchmark, SAMA-4B reaches 32.6, surpassing its Sa2VA-4B variant by a notable margin of +1.6. **(2)** SAMA sets new state-of-the-art results in video segmentation. On video benchmarks, SAMA-8B achieves 53.7 on MeViS, 77.3 on Ref-DAVIS17, and 59.1 on ReVOS—outperforming Sa2VA-8B by 2.2, 1.4, and 1.5 points, respectively. These consistent improvements highlight SAMA's enhanced grounding ability.

Table 4: Ablation studies on spatial-temporal-context aggregator and SAMA-239K dataset.

| Method | SAMA-Bench$^G$ | | | | | Video Segmentation | | |
|---|---|---|---|---|---|---|---|---|
| | mIoU | Recall | METEOR | CIDEr | CLAIR | MeViS (val_u) [9] | Ref-DAVIS [25] | ReVOS [61] |
| SAMA-4B | 0.69 | 0.55 | 0.17 | 0.65 | 0.57 | 55.4 | 74.0 | 58.8 |
| w/o STC | 0.66 (**0.03↓**) | 0.53 (**0.02↓**) | 0.15 (**0.02↓**) | 0.63 (**0.02↓**) | 0.50 (**0.07↓**) | 57.9 (**2.5↑**) | 74.5 (**0.5↑**) | 58.3 (**0.3↓**) |
| w/o SAMA-239K | 0.46 (**0.23↓**) | 0.10 (**0.45↓**) | 0.04 (**0.13↓**) | 0.02 (**0.63↓**) | 0.22 (**0.35↓**) | 56.0 (**0.6↑**) | 74.2 (**0.2↑**) | 55.6 (**3.2↓**) |

Figure 5: Visual comparisons between our SAMA and Ferret. Best viewed with zoom.

**Performance Comparisons on Chat Benchmarks in Images and Videos** To evaluate SAMA's general conversational ability beyond SAMA-Bench, we test it on standard image and video chat benchmarks. As shown in Table 3, SAMA performs robustly overall, with only a slight drop in some general chat metrics compared to Sa2VA. These results warrant contextual interpretation. First, SAMA's primary strength lies in unifying fine-grained referring, grounding, and dialogue within videos—capabilities that most grounding-focused models lack. For instance, compared to image-level grounding models like GLaMM-7B, SAMA-8B achieves significantly higher scores on shared benchmarks such as MME (1665/593 vs. 14/9), reflecting its superior dialogue ability. Second, although SAMA is mainly optimized for grounding, it still performs competitively on general conversation tasks, achieving 80.8 on MMBench, 76.2 on SEED-Bench, and 51.8 on Video-MME, which is comparable to generalist video LMMs not tailored for grounding.

### 4.3 Ablation Studies

Here, we conduct ablation studies from both model and data perspectives to thoroughly evaluate the effectiveness of the SAMA architecture and the SAMA-239K dataset. As shown in Table 4, removing the spatial-temporal-context aggregator (`w/o STC`) from SAMA-4B leads to performance drops across all metrics on SAMA-Bench$^G$, with mIoU decreasing from 0.69 to 0.66 and CLAIR from 0.57 to 0.50. More critically, training SAMA-4B without the SAMA-239K dataset (`w/o SAMA-239K`) results in a near-total collapse of grounded dialogue capabilities, with Recall plummeting to 0.10 and CIDEr to 0.02. This highlights the dataset's essential role in supporting robust referring understanding and grounded chat. Figure 5 further illustrates that SAMA, trained on SAMA-239K, exhibits substantially stronger referential understanding and temporally grounded dialogue than Ferret. However, neither the STC module nor the SAMA-239K dataset consistently improves performance on MeViS (val_u) and Ref-DAVIS. We recognize that building universally effective components and datasets for all downstream tasks is inherently challenging and plan to address these limitations in future work.

## 5 Conclusion

In this work, we introduced a comprehensive solution to equip Video Large Multimodal Models with fine-grained referential understanding and grounded dialogue capabilities. Our contributions are threefold: **1) SAMA-239K**, a large-scale video instruction dataset designed to unify referential understanding and grounding; **2) SAMA Model**, a novel Video LMM architecture that supports detailed video comprehension and segmentation; and **3) SAMA-Bench**, a curated benchmark for evaluating integrated referential understanding and grounded dialogue in videos. Extensive experiments and analysis validate the effectiveness of our dataset, model, and benchmark. We hope our work provides

a useful step toward advancing fine-grained grounding and interactive understanding in video-based multimodal systems.

**Limitations and Future Work.** SAMA has several limitations that offer opportunities for future work. Currently, it only handles box-based prompts, and expanding to other input formats like points or masks could improve its flexibility. While strong at visual grounding, its performance on general dialogue tasks still lags behind specialized conversational models, suggesting room for better reasoning integration. Like prior work, we observe that joint training on grounding and dialogue can cause interference, pointing to a need for more robust multi-task learning approaches. Finally, SAMA-239K is currently limited to short clips; scaling to longer videos would better support research on complex, long-range temporal understanding.

## Acknowledgements

This work is in part supported by National Key R&D Program of China (Grant No. 2022ZD0160103), National Natural Science Foundation of China (Grant No. 62276067), and National Natural Science Foundation of China (Grant No. 62472104).

The computations in this research were performed using the CFFF platform of Fudan University.

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

# A Appendix

## A.1 Broader Impacts

Our SAMA imposes several positive broader impacts. **1)** SAMA introduces a unified framework that enhances the capabilities of Video Large Multimodal Models (LMMs) by bridging fine-grained referential understanding and grounded, multi-turn dialogue—two areas that have traditionally been addressed separately. This integration may encourage a shift in video understanding research toward more context-aware, multimodal systems that unify perception and interaction. **2)** SAMA-239K provides a high-quality dataset designed to support joint learning of video grounding and dialogue. It offers a valuable resource for advancing interactive video reasoning and object-centric video comprehension, and may also facilitate research in grounding alignment and conversational multi-object tracking. **3)** The SAMA model serves as an early baseline for combining referring understanding with pixel-level grounding in video. It demonstrates that detailed visual grounding and referential dialogue can be effectively optimized within a single cohesive architecture, potentially guiding future designs for complex video-language tasks. **4)** SAMA-Bench addresses a key evaluation gap by introducing the first benchmark tailored to jointly assess referential video understanding and grounded dialogue. We believe that the SAMA suite—dataset, model, and benchmark—can help steer the development of future video LMMs and foster progress in multimodal human-AI interaction, where spatial-temporal grounding and communication are tightly coupled.

**Ethical Considerations.** SAMA-239K is constructed entirely from publicly available research datasets, ensuring no private or personally identifiable data are included. The released data, model, and code will be provided under a research-only license that explicitly prohibits malicious or unethical uses, such as surveillance, disinformation, or privacy violations. We encourage responsible use of SAMA to advance transparent and beneficial research in multimodal video understanding.

## A.2 Data Curation

Table 5: SAMA-239K for referential grounded video chat training.

| Dataset | Video Clips | Q&A Pairs | Object-level Descriptions |
|---|---|---|---|
| MeViS [9] | 1,087 | 10,257 | 3,550 |
| Ref-Youtube-VOS [53] | 1,976 | 15,622 | 4,929 |
| LV-VIS [58] | 2,595 | 30,943 | 14,268 |
| SAV [50] | 2,867 | 34,729 | 12,262 |
| VidSTG [74] | 6,618 | 80,745 | 31,996 |
| **SAMA-239K** | **15,143** | **172,296** | **67,005** |

**SAMA-239K.** The details of SAMA-239K are summarized in Table 5. To construct the SAMA-239K dataset, we curated and transformed several existing video understanding datasets into a unified referentially grounded video question-answering format. As illustrated in Figure 2, our automatic data generation pipeline integrates both video segmentation and video grounding datasets. For segmentation datasets with ground-truth masks, we extract object coordinates directly from the masks. For grounding datasets lacking masks, we employ HQ-SAM to generate pseudo masks. In both cases, we visualize each object by drawing a uniquely colored bounding box and assigning it a distinct tag. These annotated video frames, along with structured object information and tailored prompts, are fed into Gemini-1.5 Pro [55] to generate fine-grained object-level descriptions and multi-turn referential grounded dialogues. Below, we outline the detailed data curation process for the five primary sources contributing to our dataset: MeViS [9], Ref-YouTube-VOS [53], LV-VIS [58], SAV [50], and VidSTG [74]. MeViS and Ref-YouTube-VOS are large-scale referring video object segmentation datasets with rich pixel-level mask annotations aligned with textual expressions. For the SAV dataset, we adopt object-level referring expressions released by Sa2VA [67]. To ensure visual diversity and avoid trivial cases, we discard videos with only a single object. For the remaining videos, we uniformly sample 16 frames and use segmentation masks to identify referred objects. Each object is visualized with a uniquely colored bounding box, establishing a one-to-one mapping between object and color. These annotated frames, along with referring expressions and color-object mappings, are integrated into prompt templates and fed to Gemini, which then generates fine-grained descriptions and multi-turn dialogues that emulate natural human interaction—from basic observations to complex reasoning about object dynamics and relationships. LV-VIS [58] is a large-vocabulary video instance

segmentation dataset with pixel-level masks and category labels. For each object, its category label is included in the prompt to guide Gemini in generating category-aware descriptions. All other steps—including frame sampling, bounding box visualization, and prompt construction—follow the same procedure. VidSTG [74] is a video grounding dataset that provides bounding box annotations but lacks mask information. We uniformly sample each video at 4-frame intervals to extract JPEG frames. The frame-level bounding boxes are then used as input to HQ-SAM [24] to generate pseudo masks. The resulting data is subsequently processed following the same pipeline as the other datasets to produce training samples.

Table 6: SAMA-Bench$^G$ for video referential grounded chat evaluation.

| Dataset | Video Clips | Q&A Pairs |
|---------|------------|-----------|
| MeViS [9] | 41 | 244 |
| Ref-Youtube-VOS [53] | 131 | 756 |
| LV-VIS [58] | 150 | 1,019 |
| VidSTG [74] | 200 | 1,019 |
| SAMA-Bench$^G$ | **522** | **3,038** |

Table 7: SAMA-Bench$^C$ for video referential captioning evaluation.

| Dataset | Video Clips | Q&A Pairs |
|---------|------------|-----------|
| MeViS [9] | 41 | 117 |
| Ref-Youtube-VOS [53] | 131 | 350 |
| LV-VIS [58] | 150 | 589 |
| VidSTG [74] | 200 | 975 |
| SAMA-Bench$^C$ | **522** | **2,031** |

**SAMA-Bench.** SAMA-Bench is constructed using the same annotation pipeline as SAMA-239K, followed by rigorous automatic filtering and manual verification to remove low-quality or ambiguous samples. To ensure comprehensive evaluation across diverse video understanding scenarios, we randomly select a subset of validation videos from four datasets: MeViS, Ref-YouTube-VOS, LV-VIS, and VidSTG. As shown in Table 6, SAMA-Bench$^G$ consists of 3,038 video referential grounded chat questions, with 244 from MeViS, 756 from Ref-YouTube-VOS, 1,019 from LV-VIS, and 1,019 from VidSTG. Similarly, Table 7 summarizes SAMA-Bench$^C$, which comprises 2,031 video referential captioning questions, sourced from the same set of videos: 117 from MeViS, 350 from Ref-YouTube-VOS, 589 from LV-VIS, and 975 from VidSTG.

Table 8: Model performance under different training data proportions.

| Training Data | MME | SEED | AI2D | MMStar | SQAtest | MMBench | Video-MME |
|---------------|-----|------|------|--------|---------|---------|-----------|
| 50% mix665k | 1409 | 56.2 | 46.5 | 39.3 | 75.1 | 43.6 | 39.5 |
| 100% mix665k | 1451 | 65.7 | 57.8 | 44.5 | 78.4 | 55.8 | 41.3 |

### A.3 More Analysis

**Improving Dialogue Performance.** We investigate two approaches to improve SAMA's performance on general dialogue. We first validated the impact of expanding conversational training data. Keeping all other factors constant during training, we adjusted the proportion of the LLaVA-v1.5-mix665k dataset used in our instruction-tuning mixture. The results in Table 8 show a clear, positive correlation between the amount of dialogue data and SAMA's performance across various image chat benchmarks, confirming this is a direct and effective improvement strategy. Second, we hypothesize that a discrepancy in output granularity between general dialogue tasks and visual grounding can cause mutual interference during joint training and thus hurt dialogue performance. To probe this, we adjust the next-token prediction loss weight and observe its impact on chat results. As shown in Table 9, we found that increasing the loss weight to 1.5 led to improved performance on most chat benchmarks. This implies that an effective strategy is to re-balance the optimization objective to focus more on the holistic, dialogue-relevant output embeddings, rather than over-emphasizing grounding-specific tokens.

Table 9: Model performance under different loss weight configurations.

| Loss Weight | SAMA-Bench^G | | | | Chat Benchmark | | | | | | |
|---|---|---|---|---|---|---|---|---|---|---|---|
| | mIoU | Recall | METEOR | CIDEr | MME | SEED | AI2D | MMStar | SQAtest | MMBench | Video-MME |
| 1.00 | 67.1 | 51.7 | 14.9 | 49.7 | 1451 | 65.7 | 57.8 | 44.5 | 78.4 | 55.8 | 41.3 |
| 1.25 | 66.4 | 52.2 | 15.2 | 51.7 | 1424 | 61.4 | 52.5 | 42.8 | 76.4 | 49.7 | 39.2 |
| 1.50 | 66.9 | 51.5 | 15.0 | 52.0 | 1479 | 66.0 | 60.0 | 45.7 | 80.0 | 54.2 | 40.0 |

Table 10: Model performance under different prompt configurations.

| Method | SAMA-Bench^G | | | | SAMA-Bench^C | |
|---|---|---|---|---|---|---|
| | mIoU | Recall | METEOR | CIDEr | METEOR | CIDEr |
| Point (1) | 0.60 | 0.44 | 0.14 | 0.46 | 0.11 | 0.20 |
| Point (2) | 0.61 | 0.45 | 0.14 | 0.48 | 0.12 | 0.24 |
| Point (4) | 0.62 | 0.46 | 0.14 | 0.49 | 0.13 | 0.29 |
| Point (8) | 0.64 | 0.48 | 0.14 | 0.48 | 0.13 | 0.30 |
| Box | 0.66 | 0.49 | 0.15 | 0.50 | 0.13 | 0.30 |
| Mask | 0.65 | 0.50 | 0.14 | 0.46 | 0.13 | 0.31 |

**Prompt Sensitivity.**  To validate SAMA's flexibility, we conducted new experiments where we extended the instruction tuning to include a mixture of prompt formats. We then evaluated the model across a diverse set of inputs, including bounding boxes, masks, and a varying number of points (1, 2, 4, and 8). The results in Table 10 demonstrate that SAMA maintains strong and consistent performance across all formats.

Table 11: Model performance on SAMA Bench-G VidSTG subset.

| Method | mIoU | Recall | METEOR | CIDEr |
|---|---|---|---|---|
| Ferret13B+SAM2 | 55.8 | 36.3 | 14.3 | 17.6 |
| SAMA-1B | 61.2 | 45.3 | 15.5 | 42.6 |

**Long-Video Evaluation.**  To demonstrate SAMA's long-video capabilities, we evaluated it on the challenging VidSTG subset of SAMA Bench-G. As shown in Table 11, the results are twofold: first, SAMA-1B achieves satisfactory performance on this long-video-centric benchmark, confirming its ability to handle minute-level videos. Second, our model outperforms the powerful image-domain model, Ferret-13B+SAM2, underscoring the unique challenges of referring and grounding in the video domain.

## A.4 Visualization

Figure 6 and Figure 7 illustrate the results of SAMA on the video referential grounded chat task.

Figure 8 illustrates the results of SAMA on the video grounded description task.

Figure 9 illustrates the results of SAMA on the video referential captioning task.

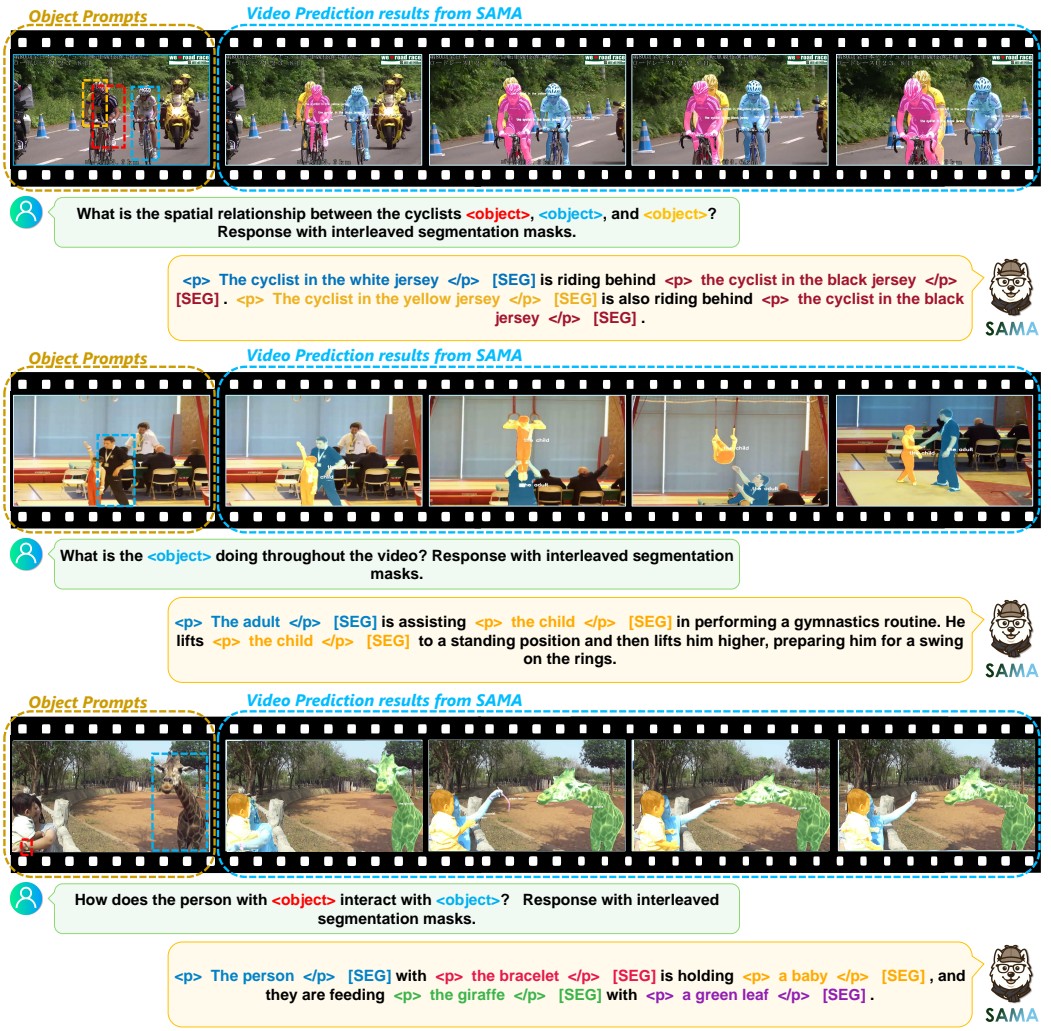

Figure 6: Visualizations of SAMA on the video referential grounded chat task. Best viewed in zoom.

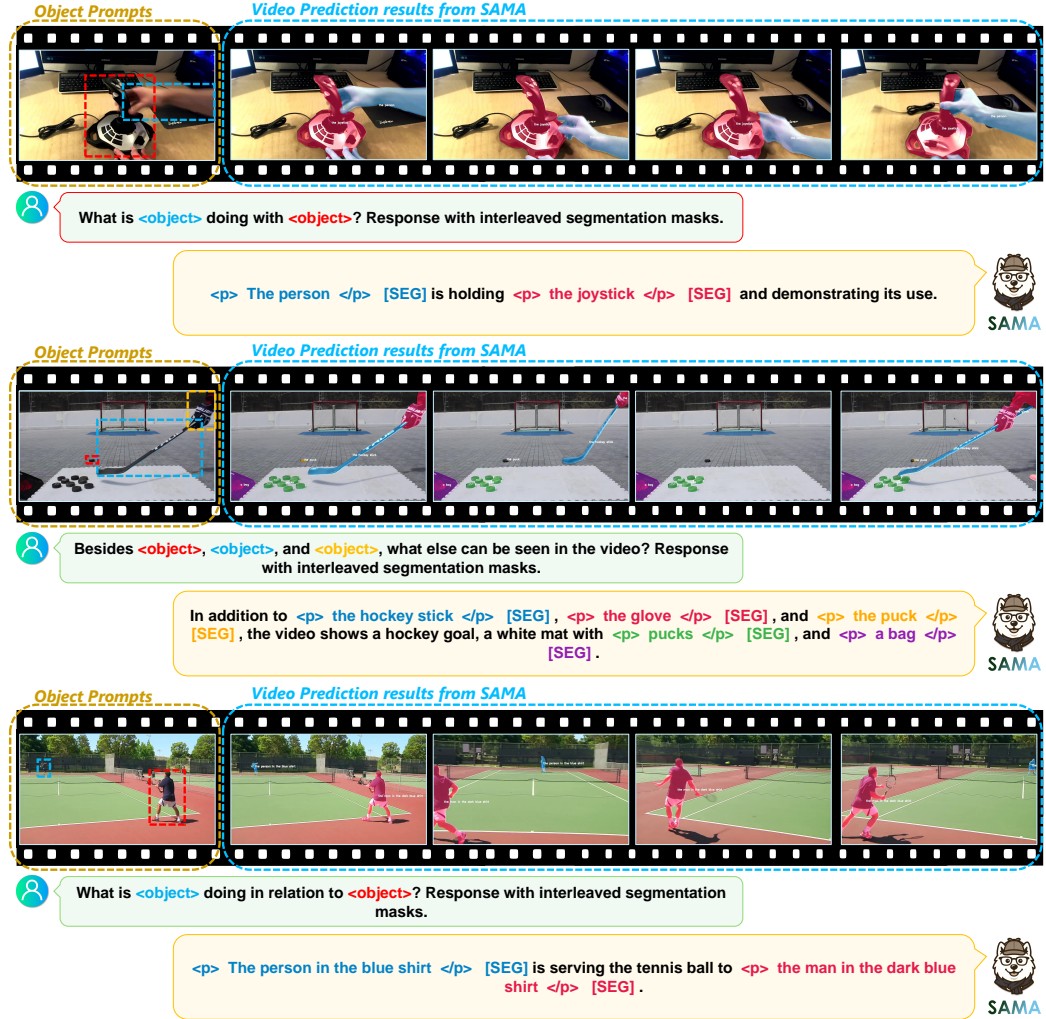

Figure 7: Visualizations of SAMA on the video referential grounded chat task. Best viewed in zoom.

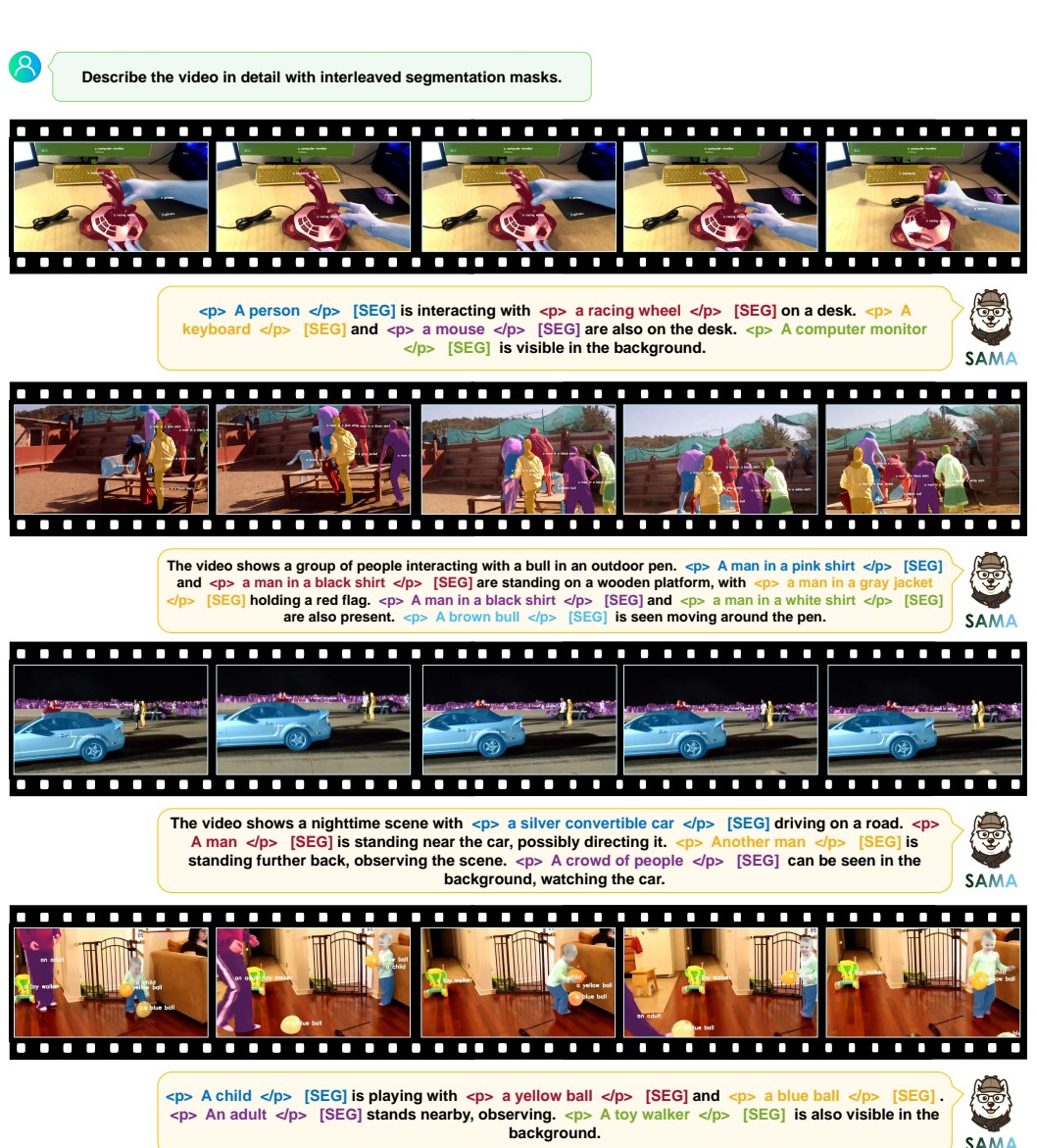

Figure 8: Visualizations of SAMA on the video grounded description task. Best viewed in zoom.

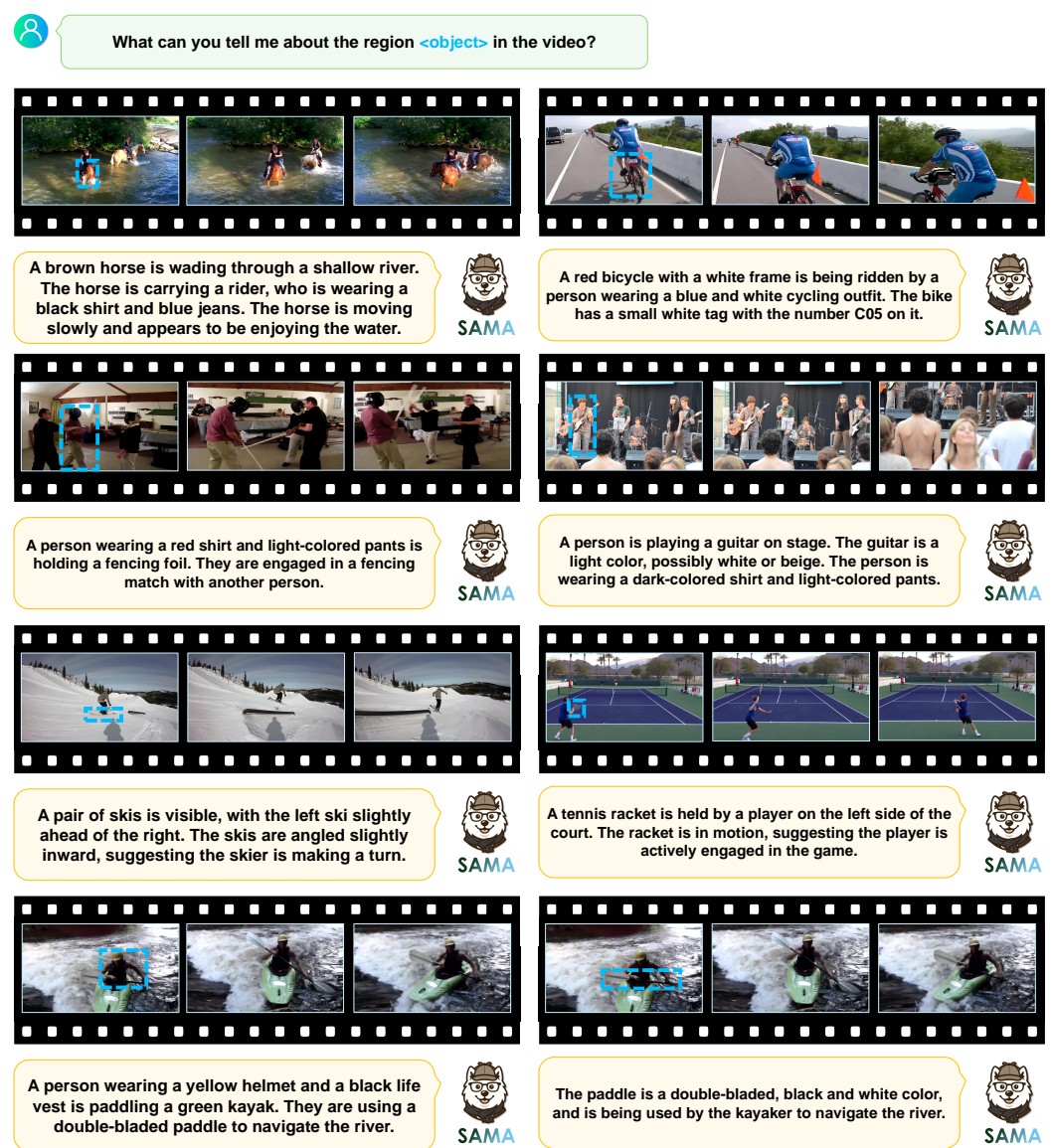

Figure 9: Visualizations of SAMA on the video referential captioning task. Best viewed in zoom.

