# OpenReview forum: "SAMA: Towards Multi-Turn Referential Grounded Video Chat with Large Language Models"
_NeurIPS.cc/2025/Conference — NeurIPS 2025 poster_

### Official Review · Reviewer_dpKE · 2025-07-01

**Clarity:** 2
**Significance:** 2
**Originality:** 2
**Rating:** 4
**Confidence:** 4

**Summary:**

This paper proposes a multimodal video understanding model that combines a STC aggregator with region-level alignment and a newly collected instruction-following dataset annotated via Gemini. The model uses advanced visual modules (e.g., InternVL 2.5 and SAM2) and introduces a unified mechanism for referential video reasoning. Experimental results on the proposed benchmark and other benches show performance improvements over existing baselines.

**Questions:**

**1. Data Verification**: Were the Gemini-generated annotations subject to any human review or post-filtering? If not, how is annotation quality ensured?

**2. Data Leakage Risk**: Since VIDSTG and some other datasets were re-annotated via SAM, how can you confirm that the validation samples (or more specifically, the objects involved in validation questions) did not appear in the training set?

**3. Benchmark Generalization**: To what extent is the benchmark task distribution influenced by the collected dataset? Could the benchmark favor models trained on your specific data distribution?

**4. Unify Module Impact**: Can you provide evidence (ablation or analysis) that the unify mechanism leads to improved reasoning or comprehension?

**Ethical Concerns:**

["NO or VERY MINOR ethics concerns only"]

**Final Justification:**

The author's response has resolved my concerns about data curation, training protocol, and comparison with similar model designs. Although the novelty issue is not fully resolved, this shall be considered as a minor problem for the LMM community.

**Limitations:**

1. There is a risk of unintentional overlap or leakage between training and evaluation sets, especially when reusing public datasets with new annotations.

2. (minor) Reliance on unverified Gemini annotations introduces uncertainty into the dataset quality. Model behavior may be driven by annotation artifacts rather than true understanding.

3. (minor) The approach depends heavily on strong pretrained components (e.g., SAM2, InternVL-2.5), which may limit scalability or deployment in less resource-rich environments.

We hope the authors can carefully think about these issues to make their paper a technically sound and useful submission.

**Quality:**

2

**Strengths And Weaknesses:**

## Strengths:

**1. Timely Task**: The focus on multimodal video understanding with open-ended, instruction-following capability is an important and emerging direction.

**2. Dataset Scale**: The authors construct a large-scale, instruction-annotated dataset using Gemini and include re-annotated samples from existing datasets.

**3. Strong Empirical Results**: The model achieves high performance on the proposed benchmark, outperforming prior baselines.


## Weaknesses:

**1. Limited Architectural Novelty**:
The model reuses familiar design elements such as RoI Align and hidden-state-as-query mechanisms, which are well-established in prior work. Although the proposed STC module bears resemblance to Q-Former, the paper does not offer sufficient discussion to clarify or justify its novelty. Moreover, the training objective and inference cost of STC are not described—Is it trained end-to-end? What are the training dynamics? Without this information, it's difficult to assess the model’s practicality or stability. As it stands, the architecture appears to be an incremental combination of known components.

**2. Dataset/Benchmark Problem**: The proposed SAMA dataset and benchmark consolidate multiple sources with auto-generated annotations, but the resulting task setup resembles existing benchmarks for referential comprehension in static images. While the authors claim SAMA *supports joint learning of video referring understanding and grounding*, the improvements on grounding benchmarks (Table 2) are marginal. Additionally, results on general video understanding tasks (Table 3) are unconvincing: SAMA even degrades performance on video-chat, indicating that its contribution to broader video understanding remains unclear and needs further validation.

**3. Underspecified Training Protocol**: Important training details are omitted, such as the training stages, and whether training was single- or multi-pass over the SFT data. This lack of clarity raises concerns about reproducibility and the risk of overfitting—especially given the Q-Former and SAM-decoder-style architecture, which is known to be prone to overfitting on limited data.

**4. Qualitative Examples Issue**: (minor) Some qualitative examples (e.g., Figure 1a), intended to illustrate the model’s reasoning ability, rely mainly on static visual cues. These outputs could likely be produced by a strong visual encoder alone, rather than reflecting meaningful referential reasoning.

---

> ### Author Rebuttal · Authors · 2025-07-31
>
> We sincerely appreciate Reviewer dpKE for their valuable feedback.
>
> **Q1: Limited Architectural Novelty.**
>
> **A1:** We thank the reviewer for this question.
> * First, to address your specific questions about the architecture: The STC module is trained end-to-end. Its output features are concatenated with other visual and text embeddings as input to the LLM. Therefore, it shares the same final optimization objectives: a token-level cross-entropy loss for next-token prediction and a pixel-level cross-entropy loss for segmentation mask generation. For training resources, please kindly refer to our response to Reviewer Ewj9 Q9.
>
> * Please allow us to clarify that our contribution is *not to propose a new, isolated technique for a pre-existing problem, but rather to leverage existing technologies to define and solve a new problem.*
>
>   Our primary objective was to address a fundamental, long-standing gap in video understanding: proving that video referring understanding and video grounding—two historically separate but deeply related tasks—could finally be unified to create a more powerful and versatile vision system.
>
>   Prior to our work, this unified task was uncharted territory: no large-scale dataset existed for joint training, no model possessed this integrated capability, and no benchmark was available for evaluation. Our work is the first comprehensive solution designed to build this field from the ground up:
>   * **We defined the task and created the foundational resources:** the SAMA-239K dataset and SAMA-Bench.
>   * **We presented the first strong baseline:** our SAMA model, which is purposefully tailored to prove this unified task is solvable.
>   * **We will fully open-source our work to accelerate community progress.**
>
>   *We genuinely argue that establishing a new research paradigm, creating the entire ecosystem of resources to support it, and presenting the first strong, reproducible baseline is a foundational and valuable contribution to the community.*
>
> **Q2: Dataset/Benchmark Problem.**
>
> **A2:** Please allow us to clarify the significance of our results and contributions.
> * **Task Setup:** While our task involves referential comprehension, it is fundamentally video-centric and demands spatio-temporal reasoning far beyond static image analysis. Our qualitative results in Figures 4-8 provide clear evidence.
> * **Grounding Improvements:** On challenging video segmentation benchmarks (Table 2), SAMA-8B establishes a new state-of-the-art, outperforming the Sa2VA with gains of +2.2 on MeViS, +1.4 on Ref-DAVIS17, and +1.5 on ReVOS. We wish to emphasize that a +2.2 gain on MeViS is significant, as this benchmark requires segmenting moving objects based on complex descriptions in dynamic environments, making it a highly challenging task. The difficulty of MeViS is further highlighted by the marginal 0.1-point gain of VISA-13B (ECCV'24) over the smaller VideoLISA-3.8B (NeurIPS'24).
> * **General Dialogue Performance:** The slight drop in chat performance compared to Sa2VA is a trade-off and a common challenge across segmentation-capable LMMs (e.g., OMG-LLaVA, LLaVA-Grounding, VideoLISA). Fine-tuning on segmentation data typically causes a decline in general dialogue capability (please see our response to Reviewer tQRK’s Q5 for further details regarding this common problem). However, it is crucial to note that while SAMA's dialogue performance is not the absolute best, it remains highly competitive and significantly surpasses that of many other segmentation-capable models, such as OMG-LLaVA and PixelLM.
>
> **Q3: Underspecified Training Protocol.**
>
> **A3:** We thank the reviewer for this question.
> * SAMA is developed through a single-stage instruction tuning process. We freeze the core parameters of the large vision encoder and the LLM. Only our proposed STC aggregator and the SAM2 mask decoder are made trainable. This allows our lightweight STC module to efficiently learn how to align spatio-temporal video features with the LLM's representation space.
> * The risk of overfitting is minimal. The number of trainable parameters is small, while our training data is a massive and diverse mixture (L249-254). This large data-to-parameter ratio ensures the model learns generalizable skills rather than memorizing the training set.
>
> **Q4: Qualitative Examples Issue.**
>
> **A4:** We thank the reviewer and agree that Figure 1(a) was a simplified example. Our full capabilities in handling complex, dynamic scenes beyond static cues are better demonstrated in Figures 4-8:
>
> * Figure 4: SAMA not only answers the main query but also accurately identifies and grounds other objects, a task where the strong Ferret baseline fails.
> * Figure 5 (second example): SAMA correctly analyzes the evolving actions of the man throughout the video.
> * Figure 7 (second example): SAMA precisely segments multiple objects in a complex scene and provides a correct behavioral description of the bull.
> * Figure 8 showcases SAMA's ability to describe the changing behaviors of various objects.
>
> **Q5: Data Verification.**
>
> **A5:** Our data underwent a rigorous verification process to ensure the quality, including automated filtering and manual verification. Please kindly refer to our detailed response to Reviewer Ewj9 Q1 due to page limitations. To summarize:
> Due to its large scale, SAMA-239K underwent a comprehensive automated post-filtering process.
> SAMA-Bench was subjected to both automated filtering and a manual human review to ensure the quality.
>
> **Q6: Data Leakage Risk.**
>
> **A6:** We can state with **absolute certainty that there is no data leakage**. We strictly adhered to the official `train/val` splits of all source datasets. Our SAMA-239K training data was built exclusively from the `train` splits, while our SAMA-Bench was built exclusively from the `val` splits, ensuring a complete and clean separation. Moreover, the validation set is carefully checked and cleaned by human annotators to ensure high-quality annotations and avoid data leakage.
>
> **Q7: Benchmark Generalization**
>
> **A7:** We can confirm the generalization of SAMA-Bench through three key facts:
> * SAMA-Bench is curated from the validation splits of four diverse public datasets (MeViS, Ref-YouTube-VOS, VidSTG, LV-VIS.), ensuring a rich variety of scenes, objects, and queries. This multi-source design makes it both the largest benchmark of its kind (e.g., 5× more questions than VideoRefer [CVPR'25], 16.5× more than VideoGLaMM [CVPR'25]) and the first to jointly evaluate referential understanding and grounding.
> * The benchmark does not unfairly favor our model. As shown in Table 1, strong baselines that were not trained on SAMA-239K (such as Ferret-13B+SAM2 and Gemini-1.5 Pro) still achieve competitive results.
> * Our model, trained on SAMA-239K, achieves state-of-the-art results on multiple external benchmarks for both image and video grounding (Table 2). This strong cross-benchmark performance would be impossible if the model had merely overfitted to our data distribution.
>
> **Q8: Unify Module Impact.**
>
> **A8:** Thanks for this question. The impact is indeed profound, but it manifests as *enabling a fundamentally new capability, rather than incrementally improving existing reasoning skills.*
>
> The primary impact of our unify mechanism is that it transforms a standard Video LMM into a model that can seamlessly perform referential dialogue and precise grounding. Before being trained with our SAMA-239K, a strong baseline like Sa2VA, despite its high general video chat scores, completely lacks this capability. Our work takes the model's ability from zero to one on this new, crucial task. This leap in capability, we argue, is the most significant form of "improved comprehension" one can achieve.
>
> This contribution is particularly vital for the video domain. While the image domain has mature unified models (e.g., Ferret, Shikra, CogVLM), this has been a long-standing gap for video, primarily due to the lack of a formal task definition and the necessary data. Our work provides the first complete solution (task definition, dataset, and benchmark) to fill this gap.
>
> Finally, while our main goal was to pioneer this new capability, we achieved it while maintaining a highly competitive level of general dialogue, far surpassing previous grounding models (e.g., PixelLM, OMG-LLaVA). We believe that perfectly balancing the injection of new skills with the preservation of all existing ones is a non-trivial challenge, and we have discussed this trade-off in our paper to inspire future work.
>
> **Q9: (minor) Reliance on unverified Gemini annotations introduces uncertainty into the dataset quality.**
>
> **A9:**
> We would like to note that using LLMs for large-scale annotation is a standard practice in this field, employed by numerous leading works such as GLaMM, VideoGLaMM, Artemis, Osprey, VideoRefer, Ferret, LLaVA-Grounding and Sa2VA.
>
> Constructing SAMA-239K was a non-trivial effort and we implemented a rigorous pipeline to ensure dataset quality. Our process included: (a) multiple rounds of iterative prompt engineering to ensure rich, correctly formatted outputs; (b) using the state-of-the-art Gemini 1.5 Pro for all annotations; (c) curating data from diverse sources to avoid dataset-specific biases; and (d) a critical two-stage data verification process.
>
> **Q10: (minor) On the reliance on strong pretrained components.**
>
> **A10:** We thank the reviewer for this question. Our reliance on strong pretrained models is a deliberate choice, consistent with the standard paradigm in the LMM era. This allows us to focus our efforts on solving a new, unsolved research problem, rather than training a foundation model from scratch.
>
> Crucially, our primary contributions, the SAMA-229K dataset and the STC aggregator, are modular and backbone-agnostic. They provide a lasting ecosystem that empowers future research and can be readily adapted to more efficient backbones for deployment in resource-constrained environments.

---

> ### Comment · Reviewer_dpKE · 2025-08-04
>
> We thank the authors for their detailed clarifications, which address most of my questions. However, the main concern remains after reviewing the entire rebuttal, including responses to other reviewers.
>
> The authors emphasize that SAMA addresses a novel task and fills a long-standing gap. However, beyond the careful curation of the SAMA dataset, it is unclear what major challenges were encountered in defining or modeling this task. In response to Reviewer tQRK, the authors note that LMMs augmented with grounding decoders (such as the LISA series) tend to solely optimize on grounding tasks, which impairs general dialogue ability—a valid and important point that we have also observed. We attribute this to the poor diversity of SFT data and tasks for early endeavors.
>
> However, this challenge has been addressed by prior work such as Chatterbox [1], which tackles multi-turn referring and grounding at the image level, also through extensive data curation and supervised fine-tuning of LMMs, and reports strong results. This appears conceptually similar to the approach taken in SAMA, and shows that SFT data with rich semantic dialogues (whether generated by LLMs or not) can solve this.
>
> Therefore, the contribution appears to primarily involve transferring a task already addressed in the image domain to the video domain, without sufficient novelty in methodology. This remains my key concern regarding the work. I look forward to the authors’ further clarification.
>
> [1] ChatterBox: Multimodal Referring and Grounding with Chain-of-Questions. Yunjie Tian *et al*. AAAI 2025. We noted that this topic has been proposed in early 2024.

---

> > ### Author Response · Authors · 2025-08-05
> >
> > We sincerely thank you for your feedback and for pointing out this relevant work.
> >
> > * While there has been significant progress in the image domain, transferring this task to the video domain remains a non-trivial challenge, due to the lack of datasets, models, and benchmarks. Our work represents a significant effort to address the following challenges:
> >   - **The First Video Dataset**: Creating the first dataset for this new task was a significant challenge. Unlike images, videos introduce unique complexities such as spatio-temporal dynamics, occlusions, and long-range dependencies. To address this, we carefully curated data from five diverse video datasets, generated pseudo-labels for VidSTG with HQ-SAM, and undertook a multi-iteration prompt engineering process. Initially, we discovered that naive prompts failed to generate high-quality dialogues with sufficient grounded objects. Through iterative prompt refinement and validation, our refined prompts successfully produced dialogues that are not only rich in grounded entities but also follow natural, from-simple-to-complex human conversational patterns. Finally, after a rigorous filtering and review process, the result of this extensive effort is SAMA-239K, the first and largest dataset in the field to simultaneously support video referential understanding and grounded chat.
> >   - **The First Baseline Model**: After creating the data, the next challenge was how to architect a model for this new task. Our initial designs, inspired by works like Artemis, relied on external tracking models to extract per-frame object embeddings and were highly inefficient. Through experiments, we developed our current, more elegant approach. While our STC aggregator is conceptually simple, it is highly effective and efficient for this new task, enabling SAMA to achieve strong results. *Pioneering works in the image domain, such as GPT4RoI and Shikra, also used simple yet effective designs to solve new problems, and we believe their impact is undeniable. We firmly believe that solving a new problem with a simple and effective method is a valuable contribution, and SAMA has the potential to be a solid first baseline for this task.*
> >   - **The First Comprehensive Benchmark**: After training our model, we faced an even greater challenge of having no benchmark to evaluate this new, unified capability. Recognizing that prior, single-dataset benchmarks were too narrow to test true generalization, we created a more demanding testbed. This involved fusing multiple diverse validation sets, filtering low-quality videos, applying our automated annotation pipeline, and finally, conducting a rigorous process of automated filtering followed by a labor-intensive manual review. Through this effort, we created a more demanding benchmark that requires a model to handle dynamic actions, rich vocabularies, and long-range scenarios simultaneously. Our evaluation shows that even strong image models like Ferret13B+SAM2 struggle on SAMA-Bench, demonstrating the unique challenges of the video domain and the necessity of our benchmark as a solid foundation for future research.
> >
> >   **In summary, the core challenge of our work was not just technical, but the "0-to-1" process of defining, building, and solving this problem for the first time in the video domain.**
> >
> > * Regarding ChatterBox, we would like to clarify two fundamental differences:
> >   - **Discrepancy in Evaluation**: There is a fundamental difference in how ChatterBox and SAMA are evaluated, making a direct comparison of their dialogue capabilities misleading. ChatterBox is evaluated on its self-constructed, open-ended MCQ (Multi-turn Chain-of-Questions) task using BERTScore. As the training set of ChatterBox, CB-300K, was specifically curated for this task, it is reasonable that the model performs well on this benchmark. In contrast, SAMA and other segmentation-capable models (OMG-LLaVA, Sa2VA, PixelLM) are evaluated on widely-accepted, multiple-choice benchmarks like MME, MMBench, SEED-Bench, AI2D and Video-MME, which are arguably more challenging. This fundamental difference in task format *(specialized open-ended QA vs. general-knowledge multiple-choice)* makes the results incomparable. It also explains why fine-tuning on referential grounded dialogue data—which encourages LLM to learn specific, open-ended answer formats—can sometimes lead to a slight performance drop on these multiple-choice benchmarks.
> >   - **Difference in Training**: SAMA is trained on a far more diverse data mixture, including videos, images, chat data, and grounding data. This makes our training task more challenging but results in a more versatile model. As a result, SAMA demonstrates strong, competitive performance across a wider range of benchmarks, including image grounding, video grounding, and dialogue.
> >
> >   We thank you again for pointing out this work, and we will cite and discuss it in our revised manuscript.

---

> > > ### Comment · Reviewer_dpKE · 2025-08-05
> > >
> > > Thanks for your detailed discussion. Your effort for curating datasets and proposing solution of this problem is appreciated. The release of this baseline will be a valuable contribution to the community. I will raise my overall rate to **4** (borderline accept). Good luck!

---

> > > > ### Author Response · Authors · 2025-08-05
> > > >
> > > > Thanks for your positive feedback. This means a lot to us. Much appreciated!

---

### Official Review · Reviewer_Ewj9 · 2025-07-01

**Clarity:** 3
**Significance:** 3
**Originality:** 2
**Rating:** 5
**Confidence:** 4

**Summary:**

Summary:
The authors introduce a new dataset, method and benchmark for multi-turn referential grounded video chat for LMMs, i.e., the task of answering questions about explicit objects and their spatio-temporal changes/interactions, in a form of a dialog. For the dataset and the benchmark, the authors build upon existing data with object-level annotations, enriched with corresponsing dialogs, derived using Gemini 1.5 pro. To solve the task, the authors propose a LMM able to handle frame-level semantic details and combine it with long-form context, to solve the task. The model achieves SoTA-level performance on the benchmark.

**Questions:**

Questions:
- What’s the duration of the videos in the dataset?
- Could the authors be more specific about “Object-level Description Generation” in Section 3.1? It’s not clear if that part is done on images or videos, and what Gemini1.5 is applied to exactly.
- It’s not very clear from the description how the explicit object handling is performed (ll179-183). Specifically, what does it mean to “performing mask pooling on the visual features F_{key} of the relevant frame using a bounding box prompt for the target object.”? Also, what do the authors mean by “inserting” object features into the language embeddings “at the position corresponding to the object reference”? Does it mean the language embedding is replaced by the visual one, or augmented in some way?
- In line 184, what is F_{agg}? It wan’t defined previously.
- How are the frames selected for long-range modelling and how some frames become the key frames?
- Judging by Table 1, the difference between 1B, 4B and 8B models is quite small. Why would that be the case?
- What are the resource requirements for training and inference of SAMA?



Suggestions:
Perhaps annotation diagram in Figure 2 with symbols (eg, F_{long}, E_{obj}, …) could make following the paper easier.

**Ethical Concerns:**

["NO or VERY MINOR ethics concerns only"]

**Final Justification:**

I thank the authors for answering my questions. This clarifies a lot. I raise my score to 5.

**Limitations:**

Discussed in the paper

**Paper Formatting Concerns:**

no concerns

**Quality:**

3

**Strengths And Weaknesses:**

Strengths:
- It’s very important to establish a benchmark for grounded video understanding
- The provided training set is highly valuable for the community
- The design of the long-context aggregation effective and efficient: processing the individual frames with Q-Former, then sliding window with another Q-former to aggregate along the temporal dimension
- Great results on video segmentation.

Weaknesses:
- At no point the authors mention using human manual involvement in the data annotation process (i.e., it’s automated by ML models). While this is acceptable for the training dataset, the benchmark samples should be annotated or at least filtered/reviewd manually.
- The authors compare to Gemini-1.5 Pro that is a generalist model and show similar performance. While one could claim that Gemini does not produce segmentations directly, one could use the bounding boxes produced by the model to derive object representations, that could be then segmented. Thus the value of this method is not very clear.

---

> ### Author Rebuttal · Authors · 2025-07-30
>
> We sincerely appreciate Reviewer Ewj9 for their insightful feedback and for recognizing our contributions.
>
> **Q1: Human manual involvement in the data annotation process.**
>
> **A1:** Our SAMA-Bench undergoes a rigorous two-stage data quality assurance process: automated filtering and manual verification. In the automated filtering stage, we employ a suite of custom scripts to programmatically clean the data. This process ensures all special markers (e.g., `<p>, </p>, [SEG]`) are correctly formatted, removes invalid tags, and verifies that each `[SEG]` token corresponds to a valid segmentation mask. In the manual verification phase, we visualize the masks and manually check their alignment with the question-answer pairs, removing any problematic data. Through these thorough checks and processes, we ensure that SAMA-Bench evaluation yields stable and reliable results without errors.
>
> **Q2: The authors compare to Gemini-1.5 Pro that is a generalist model and show similar performance. While one could claim that Gemini does not produce segmentations directly, one could use the bounding boxes produced by the model to derive object representations, that could be then segmented. Thus the value of this method is not very clear.**
>
> **A2:** While such a pipeline is possible, our SAMA model provides a solution that is superior in three key aspects: end-to-end solution, performance, and accessibility.
> * End-to-end solution: The mentioned approach in the question is a complex, multi-step pipeline requiring heavy prompt engineering and heavy post-processing to get right object-associated information. In contrast, SAMA is a single, end-to-end model that seamlessly generates masks within dialogue, which is a more robust and elegant design.
> * Performance: We have already tested this pipeline. As shown in Table 1, the "Ferret-13B + SAM2" baseline, which follows this detect-then-segment logic, is outperformed by our SAMA model in grounding accuracy (0.64 vs. 0.70 mIoU, 0.43 vs. 0.55 Recall).
> * Accessibility: Gemini-1.5 Pro is a closed-source model with a costly API. In contrast, our work provides a fully open-source and reproducible set of resources (model, data, and code), empowering the community to advance this research without relying on a black box.
>
> **Q3: What’s the duration of the videos in the dataset?**
>
> **A3:** SAMA-239K consists of 15,143 videos totaling 76.82 hours. The average video length is 18.3 seconds, with the longest video lasting 179.8 seconds. All questions in SAMA-239K are annotated by the state-of-the-art video model, Gemini-1.5 Pro, to ensure quality. **In comparison, SAMA-239K contains 14 × more videos than the VISA [a] dataset, 6 × more questions than the VideoGLaMM [b] dataset, and 10.3 × the total duration of MOSE [c] dataset. The number of questions in SAMA-Bench is 11 × greater than that in ReasonVOS [d].** Importantly, SAMA uniquely supports both referential dialogue and precise grounding.
>
> We genuinely believe the release of SAMA will significantly advance research in joint learning for video referring understanding and grounding within the community.
>
> **Q4: Could the authors be more specific about “Object-level Description Generation” in Section 3.1?**
>
> **A4:** We thank the reviewer for this question.
>
> As detailed in Appendix A.1 (L523-536), our pipeline for generating these descriptions is as follows: For a given video, we first uniformly sample 16 frames. Within these frames, each target object is visually cued with a uniquely colored bounding box. These visually annotated frames are then provided to Gemini-1.5 Pro, which is prompted to generate a detailed, object-level description based on the cued region.
>
> *We will release the complete annotation code and execution scripts for every dataset, enabling researchers to reproduce our annotation pipeline effortlessly.*
>
> **Q5: It’s not very clear from the description how the explicit object handling is performed (ll179-183). Specifically, what does it mean to “performing mask pooling on the visual features F_{key} of the relevant frame using a bounding box prompt for the target object.”? Also, what do the authors mean by “inserting” object features into the language embeddings “at the position corresponding to the object reference”? Does it mean the language embedding is replaced by the visual one, or augmented in some way?**
>
> **A5:** Thank you for this precise question. We are happy to provide a step-by-step explanation for our explicit object handling mechanism.
> * On "performing mask pooling... using a bounding box prompt": This describes how we extract a visual feature vector for a specified object region.
>   - First, a bounding box prompt (e.g., `[x_min, y_min, x_max, y_max]`) is converted into a 2D binary mask of shape `H x W`. The pixel values inside the box are set to 1, and all values outside are 0.
>   - Let the visual features of a keyframe (`F_key`) have the shape `H x W x C`.
>   - "Mask pooling" then refers to masked average pooling: we multiply the visual features `F_key` with this binary mask and then compute the average of the features only in the masked (non-zero) region.
>   - This process yields a single feature vector `E_obj` of shape `1 x C`, which serves as the compact visual representation of the target object.
> * On "inserting object features...": This describes how we inject the object's visual representation `E_obj` into the language input embedding. It is a replacement, not an augmentation.
>   - Consider the input text: `"What is the <object> doing?"`.
>   - The text is first processed by a standard tokenizer, which converts the string into a sequence of token embeddings. The special placeholder `<object>` becomes a specific token embedding in this sequence.
>   - We then replace the embedding of this `<object>` placeholder token directly with the `E_obj` vector we extracted in the previous step.
>   - The final sequence of embeddings, now containing the visual information of the object, is fed into the Large Language Model.
>
> This simple and direct replacement mechanism effectively grounds the textual reference `<object>` to its precise visual features. We will add this detailed explanation to the final version.
>
> **Q6: In line 184, what is F_{agg}? It wan’t defined previously.**
>
> **A6:** `F_agg` is already defined on line178 as the "compact yet informative representation of long-range temporal dynamics" derived from our STC aggregator. We will ensure this is crystal clear in the revision.
>
> **Q7: Frame selection for long-range and keyframes.**
>
> **A7:** We adopt a simple yet effective frame sampling strategy. We treat the first five frames of each clip as key frames​. We uniformly sample 32 frames over the full video for long‑range modeling​. Our experiments confirm that this simple method is effective enough to produce satisfactory results.
>
> **Q8: Small performance difference between 1B, 4B, and 8B models.**
>
> **A8:** We think the small performance gap between model sizes stems directly from our parameter-efficient training strategy. In our training, we freeze the core parameters of both the vision encoder and the LLM, and only train our proposed STC aggregator and the SAM2 mask decoder. This means the number of trainable parameters is small and consistent across all model variants.
>
> This deliberate design choice has several key advantages:
>
> * Efficiency: It makes the training process highly memory-efficient, allowing even the 8B model to be trained on standard hardware without risk of OOM.
> * Effective Knowledge Injection: This strategy enables our lightweight STC module to effectively align its learned temporal features with the powerful, frozen representations of any LLM, acting as a "plug-and-play" component for grounding capabilities.
> * Preservation of LLM Abilities: By only updating a small fraction of parameters, we inject the new, fine-grained grounding capability while preserving the vast knowledge of the pretrained LLM.
>
> Therefore, since the performance gains are driven by the same set of newly trained parameters in all three models, it is reasonable that the final performance differences are small. We will explore more parameter-efficient training strategies to enhance the model's performance in future work.
>
> **Q9: Resource requirements and Figure 2 suggestion.**
>
> **A9:** Thank you for this question and valuable suggestion.
>
> SAMA's training and inference were performed on 8 x A100 GPUs. The 1B, 4B, and 8B models took approximately 48 hours, 55 hours, and 82 hours to train, respectively.
>
> We also appreciate the suggestion for Figure 2 and will add the relevant symbols to the diagram in the revised version.
>
>  > [a] VISA: Reasoning Video Object Segmentation via Large Language Models. ECCV 2024
>
>  > [b] VideoGLaMM: A Large Multimodal Model for Pixel-Level Visual Grounding in Videos. CVPR 2025
>
>  > [c] MOSE: A New Dataset for Video Object Segmentation in Complex Scenes. ICCV 2023
>
>  > [d] One Token to Seg Them All: Language Instructed Reasoning Segmentation in Videos. NeurIPS 2024.

---

> ### Comment · Area_Chair_nyyo · 2025-08-05
>
> Dear Reviewer,
>
> Thank you for your dedicated efforts in reviewing this paper. We are currently in the reviewer-author discussion phase, but we have not yet seen your engagement.
>
> This year's Responsible Reviewing initiative requires all reviewers to communicate with authors during this period, emphasizing that ghosting authors is not acceptable. We kindly ask that you reply and engage with the authors.
>
> Best,
> AC

---

> ### Author Response · Authors · 2025-08-08
> **Reviewer-Author Discussion Period Ends in ONE Day**
>
> Dear Reviewer Ewj9,
>
> Thanks again for your valuable review. We have provided a detailed response to address all the concerns you raised.
> As the discussion period deadline is approaching, we would be very grateful if you could revisit our clarifications. Please let us know if any questions or ambiguities remain; we are eager to engage in further discussion and are fully prepared to answer any additional questions.
>
> We sincerely appreciate your time and effort.
>
> Warm regards,
>
> Authors

---

### Official Review · Reviewer_Dj2a · 2025-07-02

**Clarity:** 4
**Significance:** 3
**Originality:** 2
**Rating:** 4
**Confidence:** 5

**Summary:**

The paper addresses the challenge that current Video LMMs struggle with fine-grained spatio-temporal understanding, as they typically handle video referring (understanding regions) and grounding (segmenting objects) as separate tasks.

To address this, the authors present: a new dataset, SAMA-239K, for joint training, a model named SAMA, and a benchmark, SAMA-Bench.

The SAMA model architecture integrates a spatio-temporal context aggregator with SAM2 to enhance video comprehension and precise object grounding jointly.

The SAMA-239K dataset, created using an automated pipeline with Gemini-1.5 Pro, contains 239K instances of referential grounded dialogue and object descriptions to enable unified learning.

Experiments demonstrate that the SAMA model not only achieves descent performance on the new SAMA-Bench but also performs well on existing image and video referring segmentation benchmarks

**Questions:**

1. Please answer the questions and concerns in the weakness.

2. Could you provide a more detailed analysis of the "interference" phenomenon between grounding and conversational skill?  Is it a case of catastrophic forgetting of general conversational skills, or does the model learn to prioritize grounding-specific token sequences over more detailed dialogue?

3. What is your hypothesis for why the dataset and model failed to provide a benefit on MeViS and Ref-DAVIS? Do these benchmarks have distinct data characteristics that are not well-represented in your training data?

**Ethical Concerns:**

["NO or VERY MINOR ethics concerns only"]

**Final Justification:**

Thanks for the detailed rebuttal. I appreciate you running new experiments—the results showing flexibility with different prompts were very helpful, and your analysis of the task interference was insightful.

While the paper is definitely stronger now, I'm holding my score steady because some of my original concerns remain. I agree the dataset is a valuable contribution, but the model's technical novelty is still a weak point. More importantly, your own explanation for the performance on MeViS and Ref-DAVIS confirms that the model has trouble generalizing to key existing benchmarks.

Because of these limitations, the work still feels like a borderline case to me. It's a solid contribution, but the issues with novelty and generalization prevent me from raising my score at this time.

**Limitations:**

Yes

**Quality:**

3

**Strengths And Weaknesses:**

Strengths

* The paper is well written and easy to follow. The figure visualization makes it easy to comprehend the method and the purposes of the dataset.
* The paper presents a pipeline to address the problem of referentially grounded video chat, comprising a new dataset for training and testing, as well as a proposed model. The release of a new dataset may benefit the research community.
* Extensive experiments and a thorough ablation study support the paper's claim. The study confirms that both the spatio-temporal aggregator and the SAMA-239K dataset are essential for the model's high performance.

Weaknesses

1. Method.
(a) Given that this is more of a dataset paper, the proposed method is not novel; the way of disintangling spatio and temporal representation mechanisms was explored previously and is widely used in designing video models.
(b) The authors acknowledge that jointly training for grounding and dialogue can cause "interference" between the tasks, indicating a need for more robust multi-task learning strategies. This implies that the goal of solving referring and grounding simultaneously has not yet been achieved.
(b) The SAMA model's flexibility is currently limited, as it only supports box-based object prompts and cannot handle other formats, such as points or masks.
(c) The current model only operates on the objects in the videos, neglecting the action as a whole, which limits the potential for action understanding in videos.


2. Experiment.
(a) In Table 1, why is the Sa2Va performance so low? The METEROR and CIDEr score reaches 0 in benchC. Any particular reason for that?
(b) In Table 2, the ablation study reveals that the architectural and data contributions did not consistently improve performance across all tasks; specifically, on the MeViS and Ref-DAVIS benchmarks, their impact was not uniformly positive.
(c) In Table 3, the model's specialization in grounding tasks comes at a cost to its general conversational abilities. SAMA's performance shows a slight drop in some general chat metrics compared to Sa2VA. Showing a clear trade-off across different aspects.

---

> ### Author Rebuttal · Authors · 2025-07-30
>
> We sincerely appreciate your insightful, constructive and encouraging reviews.
>
> **Q1: Given that this is more of a dataset paper, the proposed method is not novel.**
>
> **A1:** Please allow us to clarify the core novelty of our work. While some architectural components are inspired by prior work, our primary contribution is not an isolated component, but a holistic, foundational solution to a new and challenging problem.
>
> Our core objective was to address a fundamental, long-standing gap in video understanding: *proving that video referring understanding and video grounding—two historically separate but deeply related tasks—could finally be unified to create a more powerful and versatile vision system.*
>
> Prior to our work, this unified task was uncharted territory: no large-scale dataset existed for joint training, no model possessed this integrated capability, and no benchmark was available for evaluation. Our work is the first comprehensive solution designed to build this field from the ground up:
>
> * **We defined the task and created the foundational resources:** Our SAMA-239K dataset and SAMA-Bench are the first of their kind, providing the essential cornerstone for all future research in this direction.
> * **We presented the first strong model:** Our SAMA serves as the crucial first baseline, validating that this unified task is not just a concept, but an achievable reality.
> * **We will fully open-source our work:** To accelerate progress, we will release all our assets—data, benchmark, code, and pretrained weights.
>
> *We genuinely argue that establishing a new research paradigm, creating the entire ecosystem of resources to support it, and presenting the first strong, reproducible baseline is a foundational and valuable contribution to the community.*
>
> **Q2: The authors acknowledge that joint training for grounding and dialogue can cause "interference" between the tasks, indicating a need for more robust multi-task learning strategies. This implies that the goal of solving referring and grounding simultaneously has not yet been achieved.**
>
> **A2:** We would like to clarify that SAMA has indeed successfully solved referring and grounding simultaneously. We have rigorously validated SAMA's unified capabilities across multiple benchmarks:
>
> * Validation on SAMA-Bench: This benchmark was specifically designed to evaluate this integrated skill.
>   - SAMA-Bench-G (3,038 questions) explicitly requires the model to answer a referring question while simultaneously providing segmentation masks for the objects in the answer, testing both skills at once.
>   - SAMA-Bench-C (2,031 questions) tests fine-grained region understanding by requiring captions for referred objects.
>
>   Through rigorous testing on both benchmarks, SAMA demonstrates state-of-the-art performance in unified referring and grounding.
> * Standard Grounding Benchmarks: Extensive evaluations on established visual grounding benchmarks, such as MeViS and ReVOS, also confirm that SAMA possesses a strong and generalizable grounding ability.
> * Furthermore, our qualitative results provide visual evidence of SAMA's unified referring and grounding capabilities.
>   - In Figure 4, SAMA not only answers the main query but also accurately identifies and grounds other objects in the background, a task where the strong Ferret baseline fails.
>   - In Figure 5 (second example), SAMA correctly analyzes the evolving actions of the man throughout the video.
>   - In Figure 7 (second example), SAMA precisely segments multiple objects in a complex scene and provides a correct behavioral description of the bull.
>   - Similarly, Figure 8 showcases SAMA's ability to describe the changing behaviors of various objects, proving its understanding is rooted in video dynamics.
>
> **Q3: The SAMA model's flexibility is currently limited, as it only supports box-based object prompts and cannot handle other formats, such as points or masks.**
>
> **A3:** Please kindly see our response to Reviewer tQRK’s Q1 due to page limitations. We have already conducted new experiments showing that our SAMA framework can be easily extended to handle point and mask prompts, demonstrating its inherent flexibility. We will include these results in the final version.
>
> **Q4: The current model only operates on the objects in the videos, neglecting the action as a whole, which limits the potential for action understanding in videos.**
>
> **A4:** SAMA already handles both. SAMA is trained on a broad data spectrum (lines 249–254), combining ChatUniV’s general video‑dialogue dataset with our SAMA‑239K object‑centric dataset. This allows the model to capture both local object details and global action semantics. To our knowledge, SAMA is the first video multimodal large model that simultaneously supports general video understanding, video referential dialogue, and video grounding.
>
> **Q5: In Table 1, why is the Sa2Va performance so low on Bench-C?**
>
> **A5:** Actually, Sa2VA does not natively support video region-level captioning. During our Bench-C evaluation, we first marked the target object in the initial frame with a colored bounding box (e.g., red) and queried the Sa2VA with questions such as `"Provide me with a description of the object within the red box."` However, we found that Sa2VA generally failed to associate the textual instruction with the visual cue, whereas Gemini responded correctly. Additionally, since Sa2VA is fine-tuned on large-scale segmentation data, it often responds with `"Sure, it is [SEG]"` instead of a descriptive text, resulting in low scores.
>
> Being trained on our SAMA-239K dataset, SAMA successfully addressed this limitation in Sa2VA and demonstrated significant improvements.
>
> **Q6: Could you provide a more detailed analysis of the "interference" phenomenon between grounding and conversational skill? Is it a case of catastrophic forgetting of general conversational skills, or does the model learn to prioritize grounding-specific token sequences over more detailed dialogue?**
>
> **A6:** We have provided a comprehensive analysis of this phenomenon in our response to Reviewer tQRK's Q5, which we kindly refer you to for the full details due to page limitations.
>
> To summarize, we agree with your assessment that the model learns to prioritize grounding-specific token sequences. As detailed in our response to Reviewer tQRK's Q5, this interference stems from discrepancies in output granularity and biases towards under-trained special tokens. Our new experiments show that this can be mitigated to some extent by re-balancing the optimization objective.
>
> Furthermore, we would like to add a crucial point of context: while there is a slight trade-off compared to dialogue-only models, SAMA's conversational ability remains highly competitive and significantly outperforms prior segmentation-capable models like OMG-LLaVA and PixelLM. Most previous reasoning segmentation models, such as LISA, VideoGLaMM, and VideoLISA, lack conversational abilities entirely.
>
> This demonstrates that for a minor trade-off in conversational skill, SAMA gains two powerful, previously non-existent capabilities in the video domain: unified referential understanding and precise visual grounding. We believe this represents a valuable and worthwhile advancement for the field.
>
> **Q7: What is your hypothesis for why the dataset and model failed to provide a benefit on MeViS and Ref-DAVIS? Do these benchmarks have distinct data characteristics that are not well-represented in your training data?**
>
> **A7:** Thank you for this highly insightful question. The performance variations on MeViS and Ref-DAVIS likely stem from a subtle but crucial misalignment between the linguistic structure of these benchmarks and our grounded dialogue task, as well as a mismatch in feature granularity.
> * MeViS and Ref-DAVIS frequently use actional phrases as referring expressions (e.g., `"a dog running in the garden"`), where the grounding target is the entire action-object entity. However, such phrases are less common in natural conversation. When adapted to our interactive, dialogue-oriented format, the same scene would be structured as:
>
>   - `Q: What is <object> doing?`
>
>   - `A: <p>The dog</p>[SEG] is running in the garden.`
>
>   In this conversational format, the `[SEG]` token is associated with the nominal reference `"The dog"`. This creates a discrepancy: the model is trained to ground the noun, whereas the benchmark evaluates its ability to ground the entire actional phrase. This misalignment is likely a primary reason for the lack of performance gain. We believe this can be addressed by refining our data generation prompts to produce grounded phrases that are more aligned with the referring segmentation style (e.g., `A: <p>The dog running in the garden</p>[SEG] is white.`).
> * On the other hand, our STC aggregator is designed to capture coarse-grained, long-range temporal context. However, benchmarks like MeViS and Ref-DAVIS demand the capture of extremely fine-grained, subtle motion details to distinguish objects. Consequently, the coarse-grained features captured by the STC may not provide a direct performance boost for these highly specialized tasks. A promising solution is to implement an adaptive keyframe sampling strategy, which would allow the model to dynamically select more frames from action-intensive moments, thus enriching the fine-grained visual input.

---

> > ### Comment · Reviewer_Dj2a · 2025-08-06
> >
> > Thanks for the detailed rebuttal. I appreciate you running new experiments. The results showing flexibility with different prompts were very helpful, and your analysis of the task interference was insightful.
> >
> > While the paper is definitely stronger now, I'm holding my score steady because some of my original concerns remain. I agree the dataset is a valuable contribution, but the model's technical novelty is still a weak point. More importantly, your own explanation for the performance on MeViS and Ref-DAVIS confirms that the model has trouble generalizing to key existing benchmarks.
> >
> > Because of these limitations, the work still feels like a borderline case to me. It's a solid contribution, but the issues with novelty and generalization prevent me from raising my score at this time.

---

> ### Comment · Area_Chair_nyyo · 2025-08-05
>
> Dear Reviewer,
>
> Thank you for your dedicated efforts in reviewing this paper. We are currently in the reviewer-author discussion phase, but we have not yet seen your engagement.
>
> This year's Responsible Reviewing initiative requires all reviewers to communicate with authors during this period, emphasizing that ghosting authors is not acceptable. We kindly ask that you reply and engage with the authors.
>
> Best,
> AC

---

> ### Author Response · Authors · 2025-08-08
>
> We sincerely appreciate your recognition of our work.
> * Regarding the concern on novelty, please allow us to clarify: unlike research that builds upon established problems, datasets, and baselines to focus solely on model improvements, our work on SAMA began with a landscape devoid of data, benchmarks, or prior models. This makes a direct comparison between SAMA and works focused on architectural refinement somewhat unfair. Also, we believe that scientific novelty is going beyond just modeling, encompassing a broad spectrum covering data, modeling, and evaluation, etc. Therefore, the novelty of SAMA is holistic, spanning multiple aspects:
>   - **Dataset-Level Novelty**: SAMA-239K is currently *the only dataset* in the field designed to support multi-turn referential grounded video dialogue. SAMA-239K consists of 15,143 videos totaling 76.82 hours. In comparison, SAMA‑239K offers 14 × more videos than VISA [ECCV'24], 6 × more questions than VideoGLaMM [CVPR'25], and 10.3 × the total duration of MOSE [ICCV'23].  Creating the first dataset for this new task was a significant challenge. To address this, we undertook a comprehensive process that involved curating data from five diverse video datasets, generating pseudo-labels for VidSTG with HQ-SAM, engineering prompts through multiple iterations, and implementing a rigorous pipeline of automatic filtering and manual verification. The result of this extensive effort is SAMA-239K, the first and largest dataset in the field to simultaneously support video referential understanding and grounded chat.
>   - **Model-Level Novelty**: SAMA is *the first model* in the field capable of referential grounded video dialogue, and it has demonstrated strong performance across a wide range of benchmarks. While our STC aggregator is conceptually simple, it is highly effective and efficient for this new task, enabling SAMA to achieve strong results. *Pioneering works in the image domain, such as GPT4RoI, Shikra, CogVLM, Osprey, LLaVA-Grounding, and LISA, also used simple yet effective designs to solve new problems, and we believe their impact is undeniable. We firmly believe that solving a new problem with a simple and effective method is a valuable contribution, and SAMA has the potential to be a solid first baseline for this task.*
>   - **Benchmark-Level Novelty**: SAMA-Bench is *the first benchmark* in the field capable of evaluating unified, referential grounded video chat. In comparison, SAMA-Bench contains 5× more questions than VideoRefer [CVPR'25], 16.5× more than VideoGLaMM [CVPR'25]，11 × greater than ReasonVOS[NeurIPS'24]. Recognizing that prior, single-dataset benchmarks were too narrow to test true generalization, we created a more demanding benchmark that requires a model to handle dynamic actions, rich vocabularies, and long-range scenarios simultaneously. Our evaluation shows that even strong image models like Ferret13B+SAM2 struggle on SAMA-Bench, demonstrating the unique challenges of the video domain and the necessity of our benchmark as a solid foundation for future research.
>
> * Regarding the concern on generalization, please allow us to clarify:
>   - As stated in our abstract, the goal of SAMA is to achieve unified, referentially grounded video interaction, not merely to propose a new model for the singular task of visual grounding. Our experiments provide solid proof that we have successfully achieved this primary objective. Furthermore, we wish to clarify the results on MeViS and Ref-DAVIS: *while training on SAMA-239K did not lead to a significant increase in performance on these two benchmarks, it also caused no significant degradation. The model's performance remains at a state-of-the-art level on both.* Crucially, in exchange for this, the model gains a completely new capability—**referential grounded dialogue**—and shows significant improvement on other benchmarks like ReVOS. Our results in Table 2 and Table 4 firmly substantiate these claims.
>   - On the other hand, we must emphasize that SAMA has undergone the most extensive evaluation of any video referring or grounding model to date. SAMA was comprehensively evaluated on 17 benchmarks. This represents 4.2x the number of benchmarks used for Artemis, 2.8x for VideoRefer, 4.2x for VideoLISA, and 4.2x for VideoGLaMM. This extensive evaluation made us realize the profound challenge of designing a single dataset and model that could achieve universal performance gains across such a diverse range of tasks. Consequently, we adopted a general, simple, yet effective architecture. Our experiments strongly demonstrate that SAMA strikes an excellent balance, achieving three key results: a novel, referential grounded video dialogue capability that did not exist before; state-of-the-art visual grounding performance; and highly competitive general chat abilities. *We firmly believe that our work has the potential to become the first solid baseline in the field.*
>
> Again, we sincerely appreciate your time and feedback.

---

### Official Review · Reviewer_tQRK · 2025-07-16

**Clarity:** 3
**Significance:** 3
**Originality:** 3
**Rating:** 4
**Confidence:** 4

**Summary:**

This paper introduces SAMA, a framework for fine-grained spatial-temporal understanding and grounded dialogue in videos.

- Dataset: SAMA-239K, a large-scale dataset with 15K videos, 172K referential grounded video-QA pairs, and 67K object-level descriptions. It includes video referring understanding, grounding, and multi-turn video chat. ​
- Model: The SAMA model integrates a spatial-temporal context aggregator and SAM2 for precise grounding. ​
- Benchmark: SAMA-Bench is composed of 522 videos with 5,067 questions.

**Questions:**

- Can SAMA be extended to handle other input formats, such as points or masks, beyond box-based prompts? ​
- What specific steps can be taken to improve SAMA's performance on general dialogue tasks? Could integrating specialized reasoning modules or leveraging additional conversational datasets help? ​
- What are the main challenges in scaling SAMA to longer videos? Are there computational or architectural limitations that need to be addressed? ​
- How does interference between grounding and dialogue tasks manifest during training? ​ Are there specific techniques or architectures that could mitigate this issue? ​
- Are there any safeguards or ethical considerations for the release of SAMA, particularly given its potential for misuse in surveillance or disinformation? ​

**Ethical Concerns:**

["NO or VERY MINOR ethics concerns only"]

**Final Justification:**

After reading the rebuttal, my concerns have been addressed. I raised my score and thanks for all the efforts.

**Limitations:**

- SAMA currently handles only box-based prompts, limiting its flexibility. ​ Expanding to other input formats like points or masks could improve usability. ​
- SAMA-239K is limited to short video clips, and scaling to longer videos would better support research on complex, long-range temporal understanding. ​

**Quality:**

3

**Strengths And Weaknesses:**

Strengths:
- The paper introduces SAMA, a unified framework that integrates fine-grained referential understanding and grounded dialogue capabilities in videos, addressing a critical gap in video-based multimodal systems. ​
- The proposed SAMA-239K dataset is a significant contribution, offering a large-scale, high-quality resource for joint learning of video referring understanding and grounding. ​
- SAMA-Bench provides a well-designed evaluation framework for assessing integrated referential understanding and grounded dialogue capabilities, filling a key evaluation gap in the field. ​
- The introduction of the spatial-temporal-context aggregator and integration with SAM2 enhances SAMA's ability to capture both fine-grained spatial details and long-range temporal dynamics, making it robust for complex video understanding tasks. ​

Weaknesses:
- SAMA currently supports only box-based prompts, which restricts its flexibility and applicability in diverse scenarios. ​ Expanding to other input formats like points or masks would improve usability. ​
- While strong in visual grounding, SAMA's performance on general dialogue tasks lags behind specialized conversational models, indicating a need for better reasoning integration. ​
- SAMA-239K is limited to short video clips, which constrains its applicability to long-range temporal understanding. ​ Scaling to longer videos would enhance its utility for complex scenarios. ​

---

> ### Author Rebuttal · Authors · 2025-07-30
>
> We sincerely thank Reviewer tQRK for the insightful feedback and for recognizing our work's contributions.
>
> **Q1: Can SAMA be extended to handle other input formats, such as points or masks, beyond box-based prompts?**
>
> **A1:** Yes, absolutely. Our framework is inherently flexible and not limited to box prompts. Initially, we focused on box prompts for two practical reasons: (1)  Point prompts can be ambiguous (e.g., clicking on a person’s shirt leaves the model unsure whether the query targets the person or the clothing). (2) Mask prompts are hard for users to supply and typically require an extra segmentation module, raising the bar for real‑world use. Box prompts offer a good balance of user-friendliness and precision.
>
> To validate SAMA's flexibility, we conducted new experiments where we extended the instruction tuning to include a mixture of prompt formats. We then evaluated the model across a diverse set of inputs, including bounding boxes, masks, and a varying number of points (1, 2, 4, and 8). The results demonstrate that SAMA maintains strong and consistent performance across all formats.
> |Prompt|SAMA Bench-G| | | |SAMA Bench-C| |
> |:----|:----|:----|:----|:----|:----|:----|
> | |mIoU|Recall|METEOR|CIDEr|METEOR|CIDEr|
> |Point(1)|0.60|0.44|0.14|0.46|0.11|0.20|
> |Point(2)|0.61|0.45|0.14|0.48|0.12|0.24|
> |Point(4)|0.62|0.46|0.14|0.49|0.13|0.29|
> |Point(8)|0.64|0.48|0.14|0.48|0.13|0.30|
> |Box|0.66|0.49|0.15|0.50|0.13|0.30|
> |Mask|0.65|0.50|0.14|0.46|0.13|0.31|
>
> **Q2: What specific steps can be taken to improve SAMA's performance on general dialogue tasks?**
>
> **A2:** Before answering this question, please allow us to briefly clarify the primary goal of our work.
>
> SAMA's core objective was not merely to advance general dialogue skills, but to address a more fundamental, long-standing gap in video understanding: *to prove that video referring understanding and video grounding—two highly related but historically separate tasks—could finally be unified to create a more powerful and versatile vision system.*
>
> Prior to our work, this unified task was undefined: no large-scale dataset existed for joint training, no model possessed this integrated capability, and no benchmark was available for evaluation. SAMA is the first comprehensive solution designed to fill this critical research vacuum, providing the definition, data, model, and benchmark for this new field. Our experiments confirm its success, demonstrating that SAMA achieves sota referential grounded video chat capabilities while maintaining highly competitive general dialogue performance.
>
> As for this question, we believe that SAMA's performance on general dialogue tasks can be enhanced via following approaches:
> * Expanding the Training Data: For the sake of a fair comparison, our SAMA was instruction-tuned on a data mixture identical to Sa2VA, which includes only 100k video‑dialogue samples from ChatUniVi. Therefore, a simple and effective method is to enrich this mixture with a larger proportion of high-quality, general conversational datasets.
> * Integrating Advanced Reasoning Methods: Recent studies [a] have shown that reinforcement learning can boost LLM performance on vision-related tasks. A promising path for SAMA is to first convert the training data to a Chain-of-Thought (CoT) format, which would teach the model to articulate its intermediate reasoning steps. Following this, we can apply a two-phase refinement process: initially, supervised fine-tuning on the CoT data would build a solid reasoning foundation, and subsequently, reinforcement learning (e.g., DPO or GRPO) could be used to optimize the model's decision-making pathways, further improving its robustness and accuracy.
>
> **Q3: SAMA-239K is limited to short video clips.**
>
> **A3:** We thank the reviewer for this question.
> * In fact, SAMA-239K is not limited to very short clips. SAMA-239K consists of 15,143 videos totaling 76.82 hours. The average video length is 18.3 seconds, with the longest video lasting 179.8 seconds. All questions in SAMA-239K are annotated by the state-of-the-art video model, Gemini, to ensure quality. **In comparison, SAMA‑239K offers 14 × more videos than VISA [b], 6 × more questions than VideoGLaMM [c], and 10.3 × the total duration of MOSE [d].** Importantly, SAMA-239K is the first dataset that uniquely supports both video referential dialogue and precise grounding.
> * More importantly, we provide a reproducible LLM-assisted annotation pipeline. This empowers the community to effortlessly generate referentially grounded dialogue data for much longer videos. *We will release all the annotation code, lowering the barrier for future research in this crucial direction.*
>
> We genuinely believe the release of SAMA will significantly advance research in joint learning for video referring understanding and grounding within the community.
>
> **Q4: What are the main challenges in scaling SAMA to longer videos?**
>
> **A4:** As we mentioned in Q3, SAMA is already capable of processing videos of up to several minutes, thanks to our training on SAMA-239K.
> However, scaling SAMA to much longer, even hour-level videos presents two primary challenges. We outline these challenges below, along with promising solutions to overcome them:
>
> * Our current approach samples five keyframes to capture fine-grained visual detail. While effective for short videos, this fixed number may be insufficient to represent the full semantic content of a very long video. A potential solution is to develop an adaptive keyframe selection strategy, which dynamically identifies the most informative or visually diverse frames based on the video's content.
> * SAMA currently samples N frames for long-range context, converting them into N tokens for the LLM. For extremely long videos, a large N could increase computational load. To mitigate this, we could employ techniques such as token clustering to condense the temporal context, or integrate memory mechanisms to efficiently manage information over extended periods.
>
> **Q5: How does interference between grounding and dialogue tasks manifest during training? Are there specific techniques or architectures that could mitigate this issue?**
>
> **A5:** This interference is indeed a common challenge for segmentation-capable LMMs. Models like Sa2VA, PixelLM, and LLaVA-Grounding all exhibit a decline in conversational ability after fine-tuning on grounding data, while others like LISA, VideoLISA, and VideoGLaMM even show a much worse performance. We hypothesize this interference stems from two primary sources:
>
> * Discrepancy in Output Granularity: General dialogue tasks require the LLM to generate *diverse and rich output embeddings*. In contrast, the typical response in segmentation training data is a terse `"Sure, it is [SEG]"`. This provides no new general dialogue knowledge and contributes little to the optimization of conversational output embeddings.
> * Imbalance in Special Token Distribution: Grounding datasets are saturated with special tokens (e.g., `<p>`,  `</p>`, `[SEG]`). Crucially, these tokens do not exist in the LLM's pre-training vocabulary and thus lack the rich, generalized representations of regular word embeddings. Their frequent appearance during fine-tuning also causes the model to over-specialize on these "weak" tokens, biasing its generation towards a rigid, non-conversational style.
>
> To explore a potential solution, we conducted a new experiment on SAMA-1B where we up-weighted the token-level cross-entropy loss for next-token prediction. We found that increasing this loss weight to 1.5 led to improved performance on most chat benchmarks. This implies that an effective mitigation strategy is to re-balance the optimization objective to focus more on the holistic, dialogue‑relevant output embeddings, rather than over-emphasizing grounding-specific tokens.
> | |SAMA Bench-G| | | |Chat Benchmark| | | | | | |
> |:----|:----|:----|:----|:----|:----|:----|:----|:----|:----|:----|:----|
> |Loss weight|mIoU|Recall|METEOR|CIDEr|MME|SEED|AI2D|MMStar|SQAtest|MMBench|Video-MME|
> |1.00|67.1|51.7|14.9|49.7|1451|65.7|57.8|44.5|78.4|55.8|41.3|
> |1.25|66.4|52.2|15.2|51.7|1424|61.4|52.5|42.8|76.4|49.7|39.2|
> |1.50|66.9|51.5|15.0|52.0|1479|66.0|60.0|45.7|80.0|54.2|40.0|
>
> Simultaneously developing grounding skills while preserving chat capabilities is a non-trivial challenge, involving a delicate interplay of data curation and training strategies. SAMA strikes a good balance: it successfully introduces the field's first referentially grounded dialogue and achieves state-of-the-art visual grounding, all while retaining highly competitive chat performance. We will discuss this trade-off in our revision to inspire future work.
>
> **Q6: Are there any safeguards or ethical considerations for the release of SAMA?**
>
> **A6:** Thanks for this question. We take ethical considerations seriously.
>
> * Data Sourcing: SAMA-239K is built upon publicly available academic datasets created for research purposes. We do not use any private or sensitive data.
> * Model Release: We will release our data, model, and code under a research-oriented license that explicitly prohibits use in malicious applications, such as surveillance, creating disinformation, or violating privacy.
> * Broader Impact Statement: We will include a comprehensive "Broader Impact and Ethical Considerations" section, discussing potential misuse and outlining our mitigation strategies to encourage responsible research and development by the community.
>
>  >[a] Visual-RFT: Visual Reinforcement Fine-Tuning. arxiv 2025
>
>  >[b] VISA: Reasoning Video Object Segmentation via Large Language Models. ECCV 2024
>
>  >[c] VideoGLaMM: A Large Multimodal Model for Pixel-Level Visual Grounding in Videos. CVPR 2025
>
>  >[d] MOSE: A New Dataset for Video Object Segmentation in Complex Scenes. ICCV 2023

---

> ### Comment · Area_Chair_nyyo · 2025-08-05
>
> Dear Reviewer,
>
> Thank you for your dedicated efforts in reviewing this paper. We are currently in the reviewer-author discussion phase, but we have not yet seen your engagement.
>
> This year's Responsible Reviewing initiative requires all reviewers to communicate with authors during this period, emphasizing that ghosting authors is not acceptable. We kindly ask that you reply and engage with the authors.
>
> Best,
> AC

---

> ### Author Response · Authors · 2025-08-08
>
> Dear Reviewer tQRK,
>
>
> Thank you for your efforts in reviewing our paper. We have prepared a detailed response addressing your questions. Please kindly have a look and let us know whether it resolves your concerns. Your further feedback is valuable to us, and we hope to resolve any remaining issues before the rebuttal period ends.
>
>
> Since our last response, we conducted additional experiments to further address your concerns. Please allow us to update you with the following new results:
> * **Performance Enhancement via Data Scaling**: We first validated the impact of expanding conversational training data. Keeping all other factors constant during training, we adjusted the proportion of the LLaVA-v1.5-mix665k dataset used in our instruction-tuning mixture. The results show a clear, positive correlation between the amount of dialogue data and SAMA's performance across various image chat benchmarks, confirming this is a direct and effective improvement strategy.
>
>
>
>
>   |Training Data|MME|SEED|AI2D|MMStar|SQAtest|MMBench|Video-MME|
>   |:----|:----|:----|:----|:----|:----|:----|:----|
>   |50% mix665k|1409|56.2|46.5|39.3|75.1|43.6|39.5|
>   |100% mix665k|1451|65.7|57.8|44.5|78.4|55.8|41.3|
>
>
>
>
> * **Robust Performance on Minute-Level Long Videos**: To demonstrate SAMA's long-video capabilities, we evaluated it on the challenging VidSTG subset of SAMA Bench-G. This subset consists of 200 videos with an average duration of 34.2 seconds and a maximum of 100.3 seconds. The results are twofold:
>   - First, SAMA-1B achieves satisfactory performance on this long-video-centric benchmark, confirming its ability to handle minute-level videos.
>   - Second, our model outperforms the powerful image-domain model, Ferret-13B+SAM2, underscoring the unique challenges of referring and grounding in the video domain.
> |Method|SAMA Bench-G (VidSTG Part)| | | |
> |:----|:----|:----|:----|:----|
> | |mIoU|Recall|METEOR|CIDEr|
> |Ferret13B+SAM2|55.8|36.3|14.3|17.6|
> |SAMA-1B|61.2|45.3|15.5|42.6|
>
>
>   Finally, scaling referential video understanding and grounding to much longer, even hour-level videos remains an open challenge in the field, which we plan to explore in our future work.
>
>
> We hope these detailed responses have addressed your concerns and sincerely look forward to your feedback.

---

> ### Author Response · Authors · 2025-08-08
> **Reviewer-Author Discussion Period Ends in ONE Day**
>
> Dear Reviewer tQRK,
>
> Thanks again for your valuable review. We have provided a detailed response to address all the concerns you raised.
> As the discussion period deadline is approaching, we would be very grateful if you could revisit our clarifications. Please let us know if any questions or ambiguities remain; we are eager to engage in further discussion and are fully prepared to answer any additional questions.
>
> We sincerely appreciate your time and effort.
>
> Warm regards,
>
> Authors

---

> ### Comment · Reviewer_tQRK · 2025-08-08
> **Thanks for the detailed response!**
>
> After reading the rebuttal, my concerns have been addressed. Will raise my score.

---

> > ### Author Response · Authors · 2025-08-09
> >
> > We are glad to hear that our response has successfully addressed your concerns! Your acknowledgement means a lot to us.
> >
> > We sincerely appreciate your positive evaluation and will include the additional results and discussions in the revision.
> >
> > Thanks again for your constructive feedback and support of our work.

---

### Decision · Program_Chairs · 2025-09-17

**Decision:**

Accept (poster)

**Comment:**

The post-rebuttal discussion resulted in a positive consensus among the reviewers. Three reviewers acknowledged that their concerns are fully resolved and consequently increased their ratings. The remaining reviewer acknowledged the paper's solid contribution on the dataset side, but still had reservations about the technical novelty of the proposed model. Despite this, the reviewer concluded that the paper is a borderline case but ultimately leans toward acceptance.

Based on these final ratings and discussions, the AC recommends acceptance of the paper.